# Environmental controls on sediment grain properties of Peruvian coastal river basins

**Camille Litty[1], Fritz Schlunegger[1], Willem Viveen[2]**

*[1] Institute of Geological Sciences, University of Bern, Baltzerstrasse 1+3, CH- 3012 Bern.*

*[2] Sección de Ingeniería de Minas e Ingeniería Geológica, Departamento de Ciencias e Ingeniería, Pontificia Universidad Católica del Perú, San Miguel, Lima, Perú.*

**ABSTRACT**

Twenty-one coastal rivers located on the western Peruvian margin were analyzed to determine the relationships between fluvial and tectonic processes and sediment grain properties. Modern gravel beds were sampled along a north-south transect on the western side of the Peruvian Andes where the rivers cross the tip of the mountain range, and at each site the long *a*-axis and the intermediate *b*-axis of about 500 pebbles were measured. Morphometric properties of each drainage basin were determined and compared against measured grain properties. Grain size data show a large scatter in the $D_{50}$, $D_{84}$, $D_{96}$ values and in the ratio between the intermediate and the long axis. We have not found any correlations between the frequency of earthquakes and the grain size pattern, which suggests that the current seismic, and likewise tectonic, regime has no major controls on the supply of material on the hillslopes and the grain size pattern in the trunk stream. However, positive correlations between water shear stresses, mean basin denudation rates, mean basin slopes and basin sizes on nearly all grain size percentiles suggest a geomorphic control where larger denudation rates operating in larger basins, and steeper basins, paired with

larger flow shear stresses, are capable of transporting more and coarser grained material.
Furthermore, we use correlations between the clasts' sphericities and transport distances to infer
a transport time control on the shape of the clasts. We thus suggest that the grain size distribution
of gravel bars and the fabric of individual clasts has dynamically adjusted to water and sediment
flux and their specific time scales.
**1. INTRODUCTION**
The size and shape of gravels bear crucial information about (i) the transport dynamics of
mountain rivers (Hjulström, 1935; Shields, 1936; Blissenbach, 1952; Koiter et al., 2013;
Whittaker et al., 2007; Duller et al., 2012; Attal et al., 2015), (ii) the mechanisms of sediment
supply and provenance (Parker, 1991; Paola et al., 1992a, b; Attal and Lavé, 2006), and (iii)
environmental conditions such as uplift and precipitation (Heller and Paola, 1992; Robinson and
Slingerland, 1998; Foreman et al., 2012; Allen et al., 2013; Foreman, 2014). The mechanisms by
which grain size and shape change from source to sink have often been studied with flume
experiments (e.g. McLaren and Bowles, 1985; Lisle et al., 1993) and numerical models (Hoey
and Ferguson, 1994). These studies have mainly been directed towards exploring the controls on
the downstream reduction in grain size of gravel beds (Schumm and Stevens, 1973; Hoey and
Fergusson, 1994; Surian, 2002; Fedele and Paola, 2007; Allen et al., 2016). In addition, it has
been proposed that the grain size distribution particularly of mountainous rivers reflect the
erosional processes at work on the bordering hillslopes. This has recently been illustrated based
on a study encompassing all major rivers in the Swiss Alps with sources in various litho-teconic
units of the Central European Alps (Litty and Schlunegger, 2017). Among the various processes,
the supply of material through landsliding (van den Berg and Schlunegger, 2012) and torrential
floods in tributary rivers (Bekaddour et al., 2013) were proposed to have the greatest influence
on the grain size distribution in these rivers (Allen et al., 2013), where tributary pulses of
sediment supply alter the caliber of the trunk stream material. Accordingly, the nature of
erosional processes on valley flanks are likely to have a measurable impact on the supply of
material to the valleys' trunk rivers, and thus on the sediment caliber in these streams.
Among the various conditions, hillslope erosion and the supply of material to the trunk stream
has been shown to mainly depend on: (i) tectonic uplift resulting in steepening of the entire
landscape (Dadson et al., 2003; Wittmann et al., 2007; Ouimet et al., 2009), (ii) earthquakes and
seismicity causing the release of large volumes of landslides (Dadson et al., 2003; McPhillips et
al., 2014), (iii) precipitation rates and patterns, controlling the streams' runoff and shear stresses
(Litty et al., 2017), and (iv) bedrock lithology where low erodibilty lithologies are sources of
larger volumes of material (Korup and Schlunegger, 2009). Because most of the bedload
material of rivers has been derived from hillslopes bordering these rivers, as mapping and grain
size analyses of modern rivers in the Swiss Alps have shown (Bekaddour et al., 2014; Litty and
Schlunegger, 2017), it is very possible that the grain size distribution of modern rivers either
reflect the seismic processes at work, or rather reveal the response to the climate conditions such
as rainfall rates and the shear stresses of rivers.
The western margin of the Peruvian Andes represents a prime example where these mechanisms
and related controls on the grain size distribution of river sediments can be explored. In
particular, this mountain belt experiences intense and frequent earthquakes (Nocquet et al., 2014)
in response to subduction of the oceanic Nazca plate beneath the continental South American
plate at least since late Jurassic times (Isacks, 1988). Therefore, it is not surprising that erosion
and the transfer of material from the hillslopes to the rivers has been considered to strongly
depend on the occurrence of earthquakes, as measured [10]Be concentrations in pebbles suggest
(McPhilipps et al., 2014). On the other hand, it has also been proposed that denudation in this
part of the Andes is controlled by the distinct N-S and E-W precipitation rate gradients. These
inferences have been made based on concentrations of in-situ cosmogenic [10]Be measured in
river-born quartz (Abbühl et al., 2011; Carretier et al., 2015; Reber et al.,in press), and on
morphometric analyses of the western Andean landscape (Montgomery et al., 2001). Because
erosion has been related to either the occurrence of earthquakes and thus to tectonic processes
(McPhillips et al., 2014) or rainfall rates (Abbühl et al., 2011; Carretier et al., 2015) and thus to
the stream's mean annual runoff (Reber et al., in press), and since hillslope erosion and the
supply of material to trunk streams is likely to influence, or at least to perturb, the caliber of the
bedload material in mountainous streams (Bekaddour et al., 2013), it is possible that the grain
size pattern in Peruvian trunk rivers reflects the ensemble of these mechanisms at work.
Here we present data on sediment grain properties from rivers situated on the western margin of
the Peruvian Andes (Figure 1A) in order to elucidate the possible effects of intrinsic factors such
as morphometric properties of the drainage basins, and extrinsic properties (runoff and seismic
activity) on sediment grain properties. To this extent, we collected grain size data from gravel
bars of each stream along the entire western Andean margin of Peru that are derived from 21,
over 700-km$^2$-large basins. Sampling sites were situated at the outlets of valleys close to the
Pacific Coast.

*1.1 Geologic and tectonic setting*
The study area is located at the transition from the Peruvian Andes to the coastal lowlands along
a transect from the cities of Trujillo in the north (8°S) to Tacna in the south (18°S). In northern
and central Peru, a flat, up-to 100 km, broad coastal forearc plain with Paleogene-Neogene and
Quaternary sediments connects to the western Cordillera. This part of the western Cordillera
consists of Cretaceous to late Miocene plutons of various compositions (diorite, but also tonalite,
granite and granodiorite) that crop out over an almost continuous 1600-km long arc that is
referred to as the Coastal Batholith (e.g. Atherton, 1984; Mukasa, 1986; Haederle and Atherton,
2002; Figure 1B). In southern Peru, the coastal plain gives way to the Coastal Cordillera that
extends far into Chile. The western Cordillera comprises the central volcanic arc region of the
Peruvian Andes with altitudes of up to 6768 m.asl, where currently active volcanoes south of
14°S of latitude are related to a steep slab subduction. On the other hand, Cenozoic volcanoes in
the central and northern Peruvian arc have been extinct since c. 11 Ma due to a flat slab
subduction, which inhibited magma upwelling from the asthenosphere (Ramos, 2010).
The bedrock of the Western Cordillera is dominated by Paleogene, Neogene and Quaternary
volcanic rocks (mainly andesitic or dacitic tuffs, and ignimbrites) originating from distinct
phases of Cenozoic volcanic activity (Vidal, 1993). These rocks rest on Mesozoic and Early
Tertiary sedimentary rocks (Figure 1B).
The local relief along the western Cordillera has been formed by deeply incising rivers that flow
perpendicular to the strike of the Andes (Schildgen et al., 2007). The morphology of the
longitudinal stream profiles is characterized by two segments separated by a distinct knickzone
(Trauerstein et al., 2013). These geomorphic features have formed through headward retreat in
response to a phase of enhanced surface uplift during the late Miocene (e.g., Schildgen et al.,
2007). Upstream of these knickzones, the streams are mainly underlain by Tertiary
volcanoclastic rocks, while farther downstream incision has disclosed the Coastal Batholith and
older meta-sedimentary units (Trauerstein et al. 2013). The upstream edges of these knickzones
also delineate the upper boundaries of the major sediment sources (Litty et al., 2017). In contrast,
little to nearly zero clastic material has been derived from the headwater reaches in the Altiplano,
where the flat landscape has experienced nearly zero erosion, as 10Be-based denudation rate
estimates (Abbühl et al., 2011) and provenance tracing have shown (Litty et al., 2017).
The tectonic conditions of the western Andean are characterized by strong N-S gradients in
Quaternary uplift, seismicity and long-term subduction processes. In particular, the coastal
segment south of 13°S and particularly south of 16°S hosts raised Quaternary marine terraces
(Regard et al., 2010), suggesting the occurrence of surface uplift at least during Quaternary
times. This is also the segment of the Andes where the Nazca plate subducts at a steep angle and
where the current seismicity implies a relatively high degree of interseismic coupling, resulting
in a high frequency of earthquakes with magnitudes M>4 (Nocquet et al., 2014). In contrast, the
northern segment of the coastal Peruvian margin hosts a coastal plain that has been subsiding
(Hampel, 2002). Also in this region, the interseismic coupling along the plate interface is low, as
revealed by the relatively low frequency of earthquake occurrence (Nocquet et al., 2014).

*1.2 Climatic setting*
The N-S-oriented, annual rainfall rates decrease from 1000 mm per year near the Equator to 0
mm along the coast in southern Peru and northern Chile (Huffman et al., 2007; Figure 1C). The
Peruvian western margin shows an E-W contrasting precipitation pattern with high annual
precipitation rates up to 800 mm on the Altiplano and c. 0 mm per year on the coast (Figure 1C).
This precipitation gradient in the western Andes is related to the position of the Intertropical
Convergence Zone (ITCZ, inset of Figure 1C) associated with an orographic effect on the eastern
side of the Andes (Bookhagen and Strecker, 2008). During austral summer (January) the center
of the ITCZ is located farther south, transferring the moisture from the Amazon tropical basin to
the Altiplano (Garreaud et al., 2009) and leading to a wet climate on the Altiplano with strong
precipitation rates. During austral winter, the Altiplano is under the influence of dry air masses
from the subsiding branch of the Hadley cell that result in a more equatorial position of the ITCZ
and in a dry persistent westerly wind with almost no precipitation on the Altiplano. Additionally,
the Andes form an orogenic barrier preventing Atlantic winds and moisture to reach the coast.
Only in northern Peru around 5°S latitude, the ocean water sufficiently warms up because of the
mixing with the tropical current derived from Ecuador, resulting in precipitation in northern
Peru. In addition, every 2 to 10 years, near to the Equator, the Pacific coast is subjected to strong
precipitation resulting in high flood variability, related to the El Nino weather phenomenon
(ENSO) (DeVries, 1987).

**2. SITE SELECTION AND METHODS**
We selected river basins between 8°S and 18°S latitude situated on the western margin of the
Peruvian Andes, because of the presence of marked N-S contrasts in precipitation rates and the
presence of strong seismic activity due to the subduction of the Nazca plate (Table 1). Only the
main river basins were selected, which were generally larger than 700 km$^2$. These basins have
recently been analyzed for $^{10}$Be-based catchment averaged denudation rates and mean annual
water fluxes (Reber et al., in press). This allows us to explore whether sediment flux, which
equals the product between $^{10}$Be-based denudation rates and basin size, has a measurable impact
on the grain size pattern. In addition, also for these streams, Reber et al. (in press) presented data
on mean annual water discharge using the records of gauging stations and the TRMM-
V6.3B43.2 precipitation dataset as basis (Huffman et al., 2007). We will use this information to
explore the controls of water shear stresses on the caliber of the bedload material (see below).
Sampling sites were situated in the main river valleys in the western Cordillera just before it
gives way to the coastal margin. We selected the downstream end of these rivers because the
grain size pattern at these sites is likely to record the ensemble of the main conditions and forces
controlling the supply of material to the trunk stream farther upstream. We randomly selected c.
five longitudinal bars where we collected our grain size dataset. Sampling sites are all accessible
along the Pan-American Highway (see Table 1 for the coordinates of the sampling sites).
Additionally, the Majes basin (marked with red color on Figure 1A), which is part of the 21
studied basins, has been sampled at five sites from upstream to downstream to explore the effects
related to the sediment transport processes for a section across the mountain belt, but along
stream (Figure 2; Table 2). The Majes basin has been chosen because of its easy accessibility in
the upstream direction and because the morphology of this basin has been analyzed in a previous
study (Steffen et al., 2010).
It has been shown that using a standard frame with fixed dimensions to assist gravel sampling
reduces user-biased selections of gravels (Marcus et al., 1995; Bunte and Abt, 2001a). In order to
reduce this bias, we substituted the frame by shooting an equal number of photos at a fixed
distance (c. 1 m) from the ground surface at each longitudinal bar. Ten photos were taken from
an approximately 10 m$^2$-large area to take potential spatial variabilities among the gravel bars
into account. From those photos, the intermediate $b$-axes and the ratio of the $b$-axes and the long
$a$-axis of around 500 randomly chosen pebbles were manually measured (Bunte and Abt, 2001b)
and processed using the software program ImageJ (Rasband, 1997). Our sample population
exceeds the minimum number of samples needed for statistically reliable estimations of grain
size distributions in gravel bars (Howard, 1993; Rice and Church, 1998).
The pebbles were characterized on the basis of their median ($D_{50}$), the $D_{84}$ and the coarse ($D_{96}$)
fractions. This means that 50%, 84% and 96% of the sampled fraction is finer grained than the
$50^{th}$, $84^{th}$ and $96^{th}$ percentile of the samples. On a gravel bar, pebbles tend to lie with their short
axis perpendicular to the surface, thus exposing their section that contains the *a-* and *b*-axes
(Bunte and Abt, 2001b). However, the principal limitation is the inability to accurately measure
the fine particles < 3 mm (see also Whittaker et al., 2010). While we cannot resolve this problem
with the techniques available, we do not expect that this adds a substantial bias in the grain size
distributions reported here as their relative contributions to the point-count results are minor (i.e.
< 5%, based on visual inspection of the digital images).
Catchment-scale morphometric parameters and characteristics, including drainage area slope
angle and slope at sampling site (Table 1), were extracted from the 90-m-resolution digital
elevation model Shuttle Radar Topography Mission (SRTM; Reuter et al., 2007). The distances
from the sample sites to the upper edge of the Western Escarpment (Trauerstein et al., 2013)
have been measured.
Because grain size patterns largely depend on water shear stresses, we explored where such
correlations exist for the Peruvian rivers. We thus computed water shear stresses $\tau$ following by
Hancock and Anderson (2002) and Litty et al. (2016), where:
$$\tau = 0.54 \rho g \left(\frac{Q}{W}\right)^{0.55} S^{0.93}$$   (1).
Here, *$\rho$=1000 kg/m$^3$* is the water density, *g* the gravitational acceleration, *Q* (*m$^3$/s*) is mean
annual water discharge that we have taken from Reber et al. ( in press), *W* (*m*) the channel width,
and $S$ (*m/m*) is the channel gradient. Channel gradients at the sampling sites were calculated
using the 90-m-resolution DEM as a basis. In addition, stream channel widths at each sampling
site and at the time of the sampling campaign (May 2015) were measured on satellite images
when available, and on field images with uncertainties of about 2 m. In addition, we have
considered the basin mean denudation rates (Reber et al., in press; Table 1) as variable because
larger denudation rates points towards a larger relative sediment flux, which in turn could
influence downstream fining rates of grain sizes (Dingle et al., 2017).
Possible covariations and correlations between grain size and/or morphometric parameters and
basin characteristics were evaluated using Pearson correlation coefficients; thus providing
corresponding r-values (Table 3) and p-values with a significance level alpha < 0.1 (Table 4).
The r-values measure the linear correlations between variables. The values range between +1 and
−1, where +1 reflects a 100% positive linear correlation, 0 reflects no linear correlation, and −1
indicates a 100% negative linear correlation (Pearson, 1895). Threshold values of > + 0.30 and <
- 0.30 were selected to assign positive and negative correlations, respectively.

**3. RESULTS**
*3.1 Grain size*
The results of the grain size measurements reveal a large variation for the *b*-axis where the
values of the $D_{50}$ range from 1.3 cm to 5.5 cm for rivers along the entire western Peruvian
margin (Figure 3h; Table 1). Likewise, $D_{84}$ values vary between 3 cm and 10.5 cm. The sizes for
the $D_{96}$ reveal the largest spread, ranging from 6 cm to 31 cm. In addition, the ratio between the
lengths of the *b*-axis and *a*-axis (sphericity ratio) varies between 0.67 and 0.74 (Figure 3i). Note
that between 15.6°S and 13.7°S, no gravel bars are encountered in the rivers where they leave the
mountain range, and only sand bars can be found. Therefore no results are exhibited for these
latitudes (Figure 3h and 3i).

*3.2 The Majes basin*
The $D_{50}$ percentile of the *b*-axis decreases from 6.2 cm at 106 km river upstream to a value of 5.2
cm at 20 km upstream for the Pacific coast (Figures 2 and 4 and Table 2). Likewise, the $D_{84}$
decreases from 19 cm to 8.7 cm, and the $D_{96}$ decreases from 31 cm to 11.6 cm (Figure 4).
Geomorphologists widely accept the notion that the downstream hydraulic geometry of alluvial
channels reflects the decrease of particle size within an equilibrated system involving stream
flow, channel gradient, sediment supply and transport (e.g. Hoey and Ferguson, 1994; Fedele and
Paola, 2007; Attal and Lavé, 2009). Sternberg (1875) formalized these relations and predicted an
exponential decline in particle size in gravel bed rivers as a consequence of abrasion and
selective transport where the gravel is transported downstream. The relation follows the form: $D_x$
$= D_0 e^{-\alpha x}$ (Sternberg, 1875). Here, the exponent $\alpha$ decreases from 0.3 for the largest percentile
(i.e., the $D_{96}$) to c. 0.1 for the $D_{50}$ (Figure 4).

*3.3 Correlations between grain sizes and morphometric properties*
Table 3 shows the Pearson correlation coefficients (r-value) between the grain sizes, the
morphometric parameters and the characteristics of the basins. As was expected, the $D_{50}$, $D_{84}$ and
$D_{96}$ all strongly correlate with each other (0.73 < r-value < 0.93), but the *b/a* ratios do not
correlate with any of the 3 percentiles (-0.1 < r-value < 0.1). The $D_{50}$ values positively (but
weakly) correlate with the sizes of the catchment area (r-value = 0.31), the distances from the
Western Escarpment (r-value = 0.35), the mean annual shear stress at the sampling site (r-value =
0.23), the denudation rates (r-value = 0.34) and the sediment fluxes (r-value = 0.42; Figure
5A).The sediment fluxes show the highest significance level; p-value = 0.05 (Table 4). The $D_{84}$
and the $D_{96}$ values correlate positively with the shear stress exerted by the water on an mean
annual basis with r-value = 0.33 and 0.39 and p-value = 0.14 and 0.08 respectively (Figure 5B
and C).
The ratio of the intermediate axis over the long axis negatively correlates with the distance from
the Western Escarpment (r-value = -0.33), but a strong and positive correlation is found with the
mean slope angles of the basins (r-value = 0.63; p-value = 0.01; Figure 5D).

**4. DISCUSSION**
4.1 CONTROLS ON GRAIN SIZE
*Downstream fining trends in the Majes basin indicate fluvial controls*
In fluvial environments, the sorting of the sediment depends on the downstream distance from its
source (Hoey and Ferguson, 1994; Kodoma, 1994; Paola and Seal, 1995). This is particularly the
case for the Majes river, where the sorting gets better in the downstream direction. In particular,
we do see an exponential downstream fining trend of the three percentiles in the Majes river
(Figure 4). This is somewhat surprising because sufficiently voluminous sediment input from
other sources may perturb any downstream fining trends in the grain size distribution (Rice and
Church, 1998). Likewise, in the Majes basin, the sediment supply from the hillslopes to the trunk
stream has occurred mainly through debris flow processes and landsliding (Steffen et al., 2010;
Margirier et al., 2015). Accordingly, while the supply of hillslope-dervied material is likely to
have been accomplished by mass wasting processes, the evacuation and transport of this
sediment down to the Pacific Ocean has predominantly occurred through fluvial transport, as the
exponential downstream fining of the grains implies.

*Absence of gravels in rivers between 15.6°S and 13.7 °S*
In the rivers located between 15.6°S and 13.7°S, no gravel bars are encountered where these
rives leave the mountain range, and only sand bars can be found. This suggests that the transition
from a gravel- to a sand-covered bed, i.e. the gravel front, is located along a more upstream reach
of these rivers. This transition is generally rapid (Dingle et al., 2017) and often associated with a
break in slope (Knighton, 1999). The gravel-sand transition has been interpreted to be controlled
by either the elevation of the local base level, an excess of sand supply, and breakdown of fine
gravels by abrasion (Dingle et al., 2017), or a combination of these parameters (Knighton, 1999).
In our case, these rivers do not show any particular differences compared to the other rivers
where coastal gravel bars have been found. In particular, there is no particular evidence why
preferential breakdown of gravels along these rivers should be more efficient than in rivers
farther north and south because the upstream morphometry and bedrock geology is similar. The
other explanation would be an excess of sand supplied to these rivers. However, available
information and geological maps do not display any major differences in bedrock lithologies
along strike (Figure 1B), but we note that the resolution of the geological map does not provide
enough detail about the weathering of the bedrock or the amount of regolith, which could be a
source of sand. However, these rivers are situated in the segment where the buoyant Nazca Ridge
is being subducted beneath the South American continental plate (Figure 3), which resulted in an
uplift pulse of the forearc during Pliocene-Quaternary times, accompanied by enhanced erosion
on the surface and at interface between the subducting and the hangingwall plate through
tectonic shear (Hampel, 2002; Hampel et al., 2004). These effects are generally recorded in the
morphology and sedimentary facies of the forearc (Hampel, 2002). Additionally, based on a
detailed morphometric analysis of the region, Wipf et al. (2008) showed that this coastal uplift
has rerouted and deflected the rivers in this area and has lengthened the downstream end of these
rivers. It is thus possible that these tectonically-driven mechanisms caused the gravel front to
step back farther into the mountain range, with the effect that the downstream terminations of
these rivers only display sand bars. But we note that this interpretation warrants further detailed
investigations, which includes a down-stream survey of the sediments in these rivers from the
headwaters to the site where they discharge into the Pacific Ocean (similar to the analyses made
along the Majes river, please see above).

*Grain size and earthquake frequency*
Landslides and debris flows represent the main processes of hillslope erosion and the main
source of sediment in tectonically active orogens (Hovius et al., 1997; Korup et al., 2011). They
are generally associated with triggers such as earthquakes and generally supply coarse and
voluminous sediments to the trunk rivers (Dadson et al., 2003; McPhillips et al., 2014). In that
sense we would infer a positive correlation between the frequency of large earthquakes and the
grain size where an increase of earthquake frequency would induce an increase of landslide
occurrence, thereby supplying coarser grained sediment from the hillslopes to the rivers.
However, no correlation has been found between the seismicity and the grain size data when
looking at the number of recorded historical earthquakes (Figure 3). We then infer that seismic
activity and particularly the subduction mechanisms do not exert a measurable control on the
grain size in the rivers of the western Peruvian Andes. Nevertheless, we do consider that the lack
of gravels in rivers where the subduction of the buoyant Nazca ridge has caused uplift of the
hangingwall plate was explained by a tectonic driving force (see section above). In particular,
since this uplift caused a re-routing of these rivers (Wipf et al., 2006) and thus a lengthening of
the river courses, the gravel front might have stepped back relative to the river mouth into the
Pacific Ocean, as we have noted above.

*Supply control on the grain size pattern*
Because we have found positive correlations between the $D_{50}$ and the basins scale properties
(basin area, mean basin slope, mean basin denudation rates, water shear stresses, sediment
fluxes), we infer that the mean grain size reflects the ensemble of a complex pattern of erosional
and sediment transport processes operating in the Peruvian basins. In particular, the positive
correlation between the size of the $D_{50}$, the basin averaged denudation rate and the morphometry
of these basins leads us to propose that environmental factors exert a major control on the pattern
of the $D_{50}$ encountered for the rivers in western Peru. In this context, it is very likely that the bulk
supply of hillslope-derived sediment to the trunk stream increases with larger basin size, mean
basin slope and basin-averaged denudation rate, as the recent study by Reber et al. (in press) has
revealed. Furthermore, while tectonic processes such as earthquake frequencies have no
measurable impact on the grain size pattern, as we have outlined above, we consider it more
likely that hillslope processes occurred in response to strong precipitation events, as suggested
Bekaddour et al. (2014) and as recently shown by the devastating mudflows and floods in coastal
Peru (March 2017) due to an El Niño event. The consequence is that higher denudation rates, and
larger basins, result in a larger sediment flux in the trunk stream, which in turn yields an increase
in the scale at which transport and deposition of material occurs (Armitage et al., 2011). Related
mechanisms are likely to shift gravel fronts in rivers towards more distal sites, which could
positively influence the mean grain size percentile of the trunk rivers in the sense that the
material will coarsen.
We note that following the results from the Majes basin, we would expect a decrease in the size
of the $D_{50}$ for larger basins and larger distances from the uppermost edge of the Western
Escarpment, because of larger transport distances and thus a higher impact related to any
downstream fining trends. While these mechanisms, i.e, fining trends of all percentiles, are likely
to be observed at the scale of individual basins, we do not consider that transport distance alone
is capable of explaining the $D_{50}$ pattern in rivers at the scale of the entire western Andean margin
of Peru. In particular, the fining rate not only depends on the abrasion (Dingle et al., 2017) and
the selective entrainment processes upon transport (Ashword and Ferguson, 1989), but also on
the rate at which sediment is supplied to the rivers (e.g. McLaren, 1981; McLaren and Bowles,
1985). Particularly, in basins where the rate of hillslope-derived supply of sediment from the
hillslopes to the trunk stream is large, the overall downstream fining rate of the material is
expected to be less, because lateral sediment pulses are likely to cause the grain size fraction to
increase. This has been exemplified for modern examples in the Swiss Alps (Bekaddour et al.,
2013) and for the Pisco river in Peru (Litty et al., 2016), where fining rates of modern stream
sediment, which record low denudation rates (Bekaddour et al., 2014), are greater than those of
Pleistocene fluvial terraces, which record fast paleo-denudation rates (Bekaddour et al., 2014).
Support for this interpretation is also provided by the positive correlation between the $D_{50}$ and
the mean basin denudation rate, where larger hillslope-derived material is likely to increase the
overall sediment flux within the rivers. The consequence is a downstream shift of the gravel front
and thus of the larger size fraction of the material, as we have interpreted above.

*Hydrological control on the grain size distribution*
Hydrodynamic conditions of rivers influence the grain size upon entrainment, transport, and
deposition (Hjulström, 1935; Komar and Miller 1973; Surian, 2002). In this sense, rivers with
larger shear stresses are capable of transporting larger clasts. Accordingly, at equilibrium
conditions, we expect a correlation between the grain-size distributions and the shear stresses
exerted by the water at our surveying sites, because greater flow strengths are required to entrain
the coarser fractions of the material that make up the river beds (e.g., Ferguson et al. 1989;
Komar and Shih 1992). This is the case in our study where the grain sizes correlate with the
shear stress values. Interestingly, the correlation coefficients between the shear stress and the
grain size percentiles increase from 0.23 for the $D_{50}$ to 0.33 for the $D_{84}$ and to 0.39 for the $D_{96}$.
This suggests that the shear stress exerted by mean annual water flows has a greater impact on
the coarse fractions than on the fine fractions of the stream sediments. While we cannot fully
explain why the larger percentiles reveal a better correlation with shear stresses of mean annual
flow conditions with the available dataset, we do infer a hydraulic control on grain size
distribution of the Peruvian rivers.

4.2 TRANSPORT DISTANCE AND SLOPE ANGLE CONTROLS ON SPHERICITY
We consider a control of the transport distance on the sphericity of the pebbles. We indeed see a
negative correlation between the sphericity and the distance from the Western Escarpment where
the major sediment sources are situated, as provenance tracing investigations have shown (Litty
et al., 2017). This suggests a decrease of the sphericity with a larger transport distance. As
particles are transported over longer distances, we actually would expect abrasion (Dingle et al.,
2017) to equalize the length of the three axes, thus making a particle more spherical. While this
concept is likely to be valid for pebbles with a homogenous fabric, it likely fails to describe the
abrasion and break-down of material with an inherited planar geologic fabric (such as gneisses
and sediments). Indeed, pebbles flatten as effects of abrasion and 3D heterogeneities of bedrock
that becomes more obvious with time and transport distance (Sneed and Folk, 1958). We note
that this is only valid if we assume a linear correlation between river length and transport time.
The reincorporation of previously abraded gravels from earlier erosion and multiple transport
cycles of clasts that were temporarily stored in cut-and-fill terrace sequences, as e.g., put forward
by Bekaddour et al. (2014) in their study about cut-and-fill terraces in the Pisco valley at c.
13.7°S latitude, would positively contribute to this effect upon increasing the time scale of
sediment evacuation.
Additionally, we consider a control of the mean catchment slope on the sphericity of the pebbles,
where correlations are positive, i.e. the steeper a basin the rounder the pebbles (Figure 5). We do
not consider that this pattern is due to differences in exposed bedrock in the hinterland because
the litho-tectonic architecture is fairly constant along the entire Peruvian margin (Figure 1).
Instead, the observations point toward the same control mechanisms on the pebble sphericity as
noted above. Steeper slope angles are most likely associated with faster denudation rates as the
Peruvian study by Reber et al. (in press) has shown. Accordingly, we infer a shorter transport
distance of the material and thus a shorter time scale of transport compared to the evacuation
time in long and less steep rivers. Similar to what we have noted above, we see the positive
correlation between mean hillslope angle and the sphericity of pebbles as a very likely
consequence of shorter transport times in steeper basins, but we note that this hypothesis needs to
be confirmed by detailed real-time surveys of material transport from sources down to the end of
these rivers.

**Conclusion**
We have conducted a grain size analysis of gravel bars in all major rivers that are situated on the
western Andean margin of Peru where they leave the mountain belt. We have not found any
correlations to the current seismic regimes, where a larger seismicity is expected to increase the
supply of coarse-grained material. Instead, we found positive correlations between water shears
stresses, mean basin denudation rates, mean basin slopes and basin sizes on nearly all grain size
percentiles. We interpret these results as the combined effect of various geomorphic conditions
where larger denudation rates operating in larger basins, and steeper slopes, paired with larger
flow shear stresses, are capable of transporting more and coarser-grained material. Furthermore,
we unravel a transport time control on the shape of the clasts where steeper slopes and smaller
basins (i.e., shorter distances to the edge of the Western Escarpment) are anticipated to shorten
the residence time of the clasts in the system, thereby yielding more spherical clasts. In
particular, longer residence times would allow abrasion to be more selective because of a planar
lithologic fabric of most of the clasts, which in turn, would cause clasts to flatten upon longer
exposure towards abrasion. This suggest that the ensemble of erosional and sediment transport
processes have reached an equilibrium at the scale of individual clasts, but also at the reach scale
of rivers where the sedimentary architecture and the clast fabric of the channel fill has
dynamically adjusted to water and sediment flux and their specific time scales. Accordingly, we
see the western Peruvian margin as ideal laboratory to analyze the relationships between
sediment supply and water runoff on the grain size pattern of the bedload, and we propose that
the bedload caliber of these streams has reached an equilibrium to environmental conditions
including water discharge, sediment flux and channel geometries.

**ACKNOWLEDGMENTS**
This project is funded by the Swiss National Science Foundation (project Number 137516).

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

**FIGURES AND TABLES CAPTIONS**

**Table 1:** Location of the sampling sites with the altitude in meters above sea level. The table also
displays grain size results together with the rivers' and basins' properties and hydrological
properties.

**Table 2:** Location of the sampling sites in the Majes basin and grain size results in the Majes
basin.

**Table 3:** Results of the statistical investigations, illustrated here as correlation matrix of the r-
values. The valuess in bold show significant correlation between the grain size data and the
different catchment scale properties.

**Table 4:** Results of the statistical investigations, illustrated here as correlation matrix of the p-
values.  The values in bold have a significance level alpha $< 0.1$

**Figure 1: A:** Map of the studied basins showing the sampling sites and the western escarpment
(western escarpment modified after Trauerstein et al., 2013). The southern and northern group of
basins represent catchments displaying differences in terms of their sizes and relationships with
grain sizes (see Results)  **B:** Geological map of the western Peruvian Andes. **C:** Map of the
precipitation rates showing the spatial extend of the ITCZ, modified after Huffman et al., 2007.

**Figure 2:** Geological map of the Majes basin overlain by the precipitation pattern (Precipitation
data from Steffen et al., 2010., where the black dashed lines show precipitation rates (mm/yr).
GS1 to GS5 represent sites where grain size data has been collected. The right corner shows the
Majes river long profile.

**Figure 3:** Topography of subducting Nazca plate, where slab depth data has been extracted from
earthquake.usgs.gov/data/slab/. This N-S projection also illustrates: a) tectonic lineaments such
as submarine ridges and MFZ: Mendaña Fracture Zone; NFZ: Nazca Fracture Zone; b) Holocene
Volcanoes; c) Earthquake data, taken from earthquake.usgs.gov/earthquakes/search/; number of
earthquakes M>4 within 30 km radius window. d) Coastal elevation. The data has been extracted
from a 20 km-wide swath prole along the coast. The three lines represent maximum, mean and
minimum elevations within the selected swath; e) Catchment averaged denudation rates have
been corrected for quartz contents (Reber et al. in press); f) Mean annual precipitation rates
(Reber et al., in press); g) Mean annual water discharge (Reber et al., in press); h) Grain size
results for the intermediate (b)-axis of the pebbles in the rivers from north to south at the
sampling sites presented in Figure 1; i) Ratio between the intermediate axis and the long (a)-axis
(modied after Reber et al., in press).

**Figure 4:** Grain size results along the Majes River.

**Figure 5:** Correlations between the grain size data and the river parameters. **A:** $D_{50}$ versus
sediment fluxes. **B:** $D_{84}$ versus shear stress exerted by the water. **C:** $D_{96}$ versus shear stress
exerted by the water. **D:** Ratio b/a versus mean catchment slope.

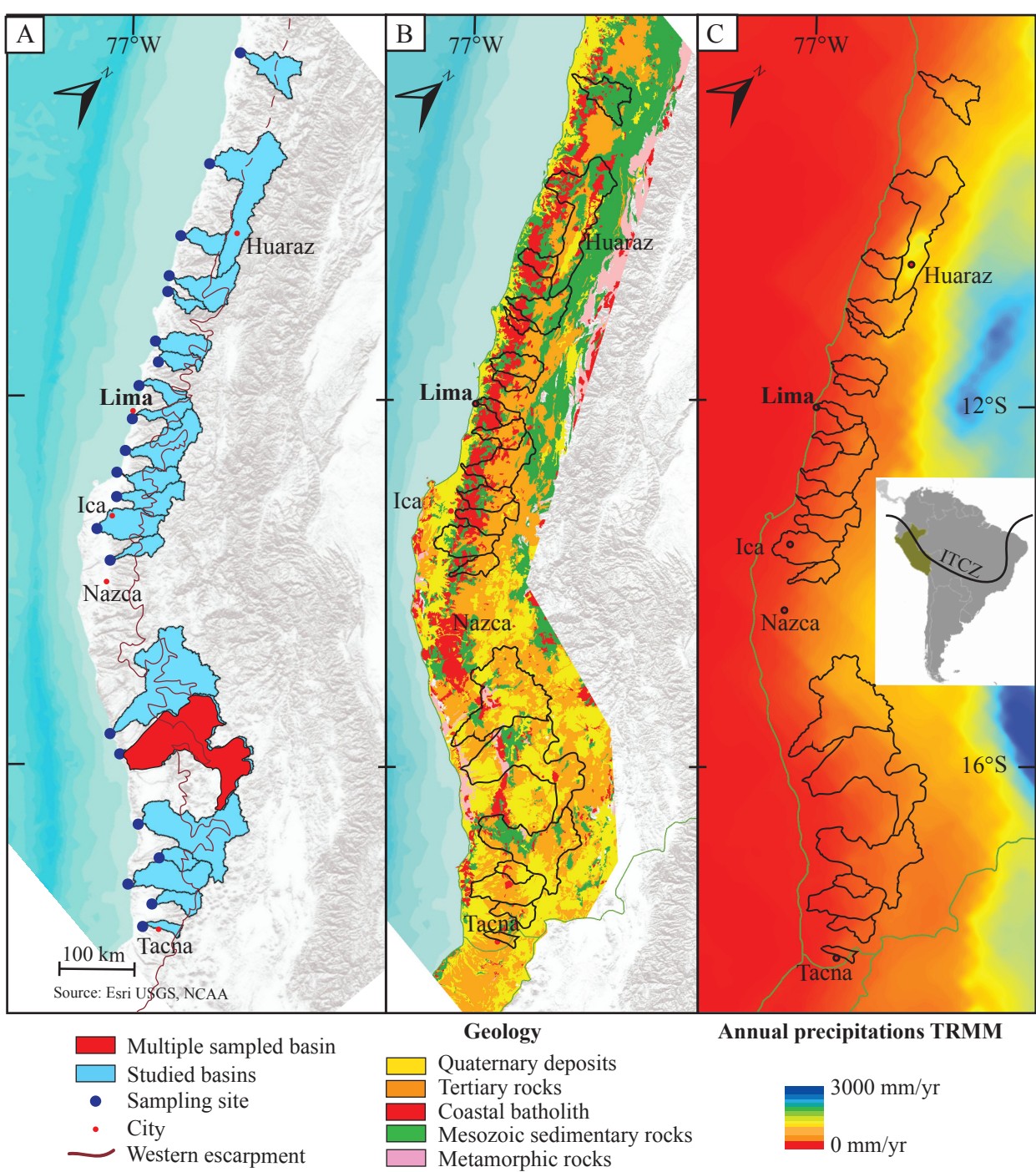

**Figure 1: A:** Map of the studied basins showing the sampling sites and the western escarpment (western escarpment modified after Trauerstein et al., 2013). **B:** Geological map of the western Peruvian Andes. **C:** Map of the precipitation rates showing the spatial extend of the ITCZ, modified after Huffman et al., 2007.)

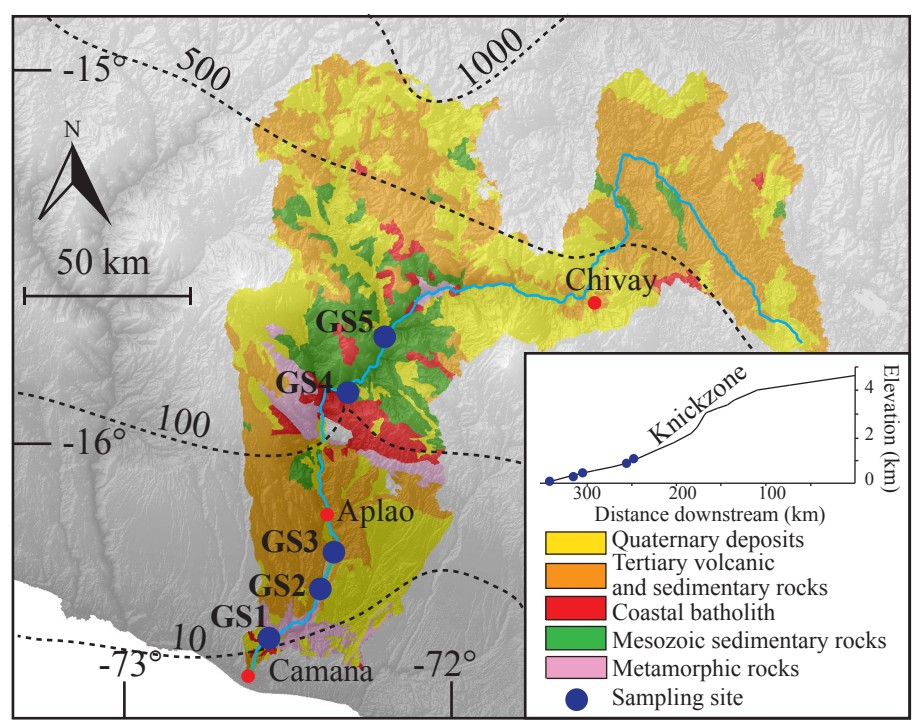

**Figure 2:** Geological map of the Majes basin overlain by the precipitation pattern (Precipitation data from Steffen et al., 2010., where the black dashed lines show precipitation rates (mm/yr). GS1 to GS5 represent sites where grain size data has been collected. The right corner shows the Majes river long profile.

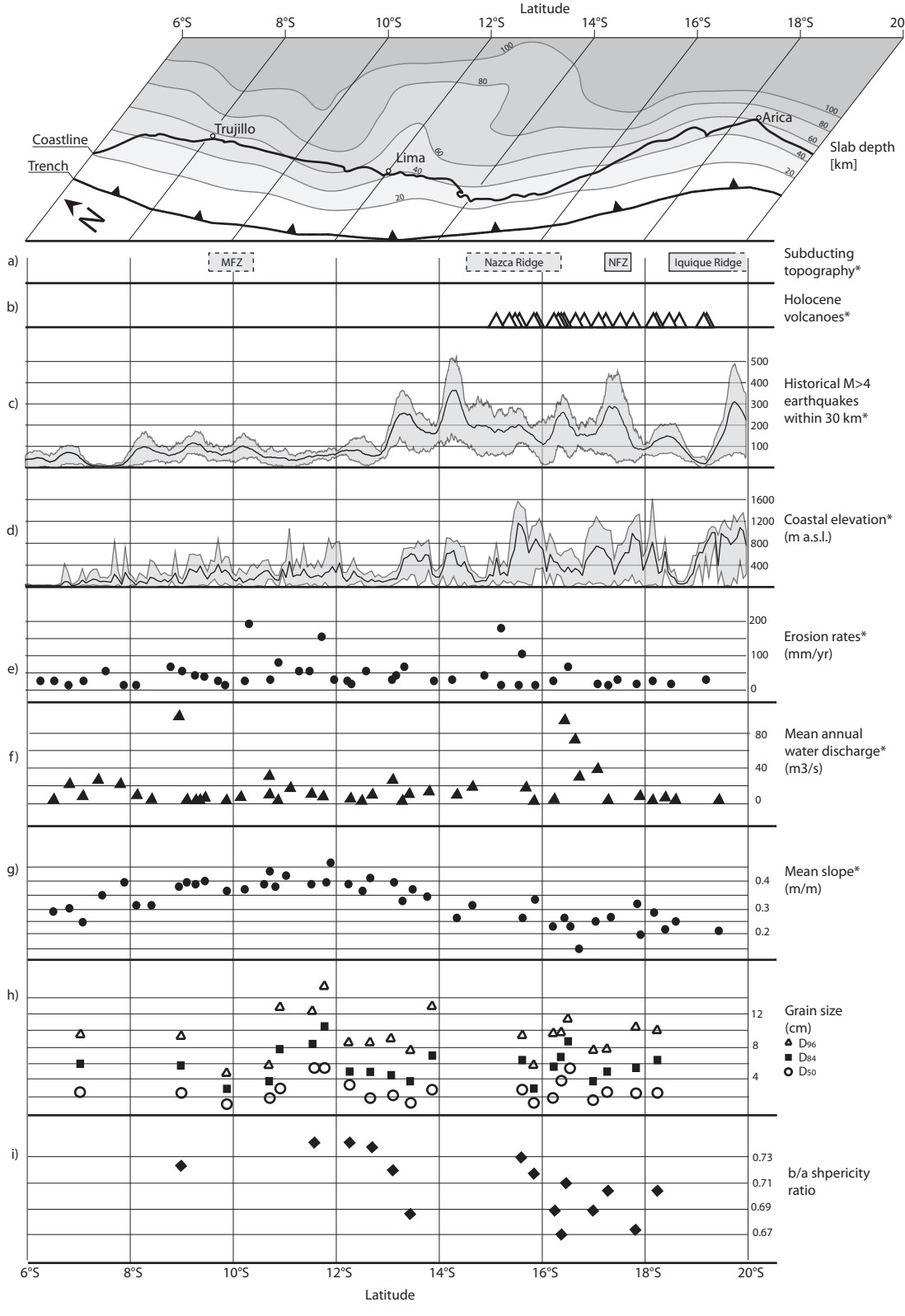

Figure 3: Topography of subducting Nazca plate, where slab depth data has been extracted from earthquake.usgs.gov/data/slab/. This N-S projection also illustrates: a) tectonic lineaments such as submarine ridges and MFZ: Mendaña Fracture Zone; NFZ: Nazca Fracture Zone; b) Holocene Volcanoes; c) Earthquake data, taken from earthquake.usgs.gov/earthquakes/search/; number of earthquakes M>4 within 30 km radius window; d) Coastal elevation. The data has been extracted from a 20 km-wide swath profile along the coast. The three lines represent maximum, mean and minimum elevations within the selected swath; e) Catchment averaged denudation rates have been corrected for quartz contents; f) Mean annual water discharge; g) Mean basin slope. h) Grain size results for the intermediate (b)-axis of the pebbles in the streams from north to south at the sampling sites presented in Figure 1; i) Ratio between the intermediate axis and the long (a)-axis (modified after Reber et al., in press).

* Data from Reber et al., in Press

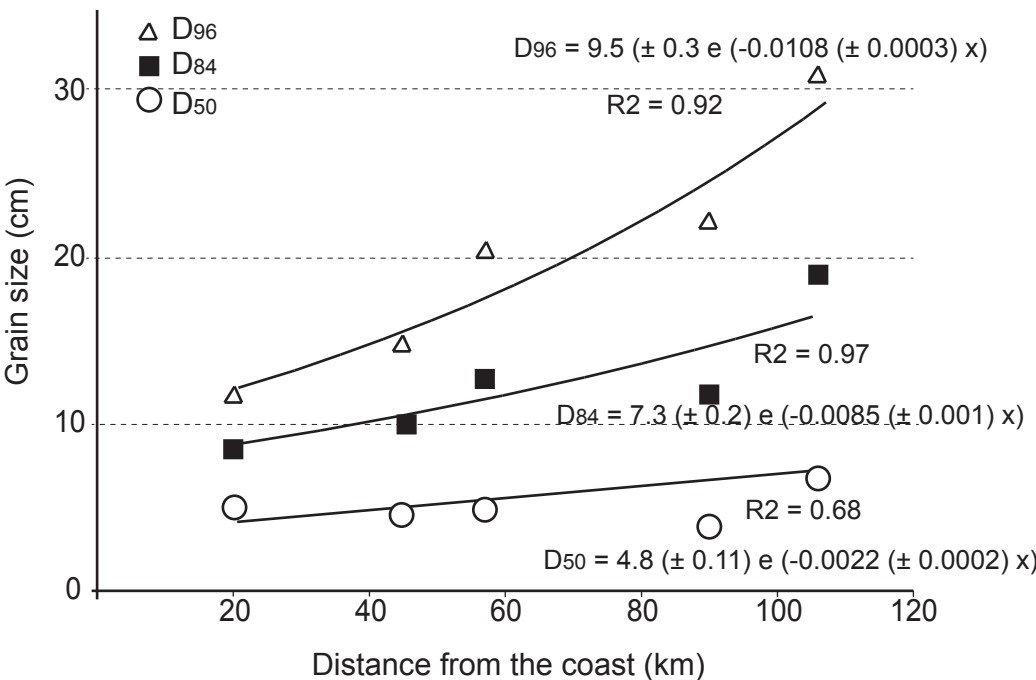

**Figure 4:** Grain size results along the Majes River.

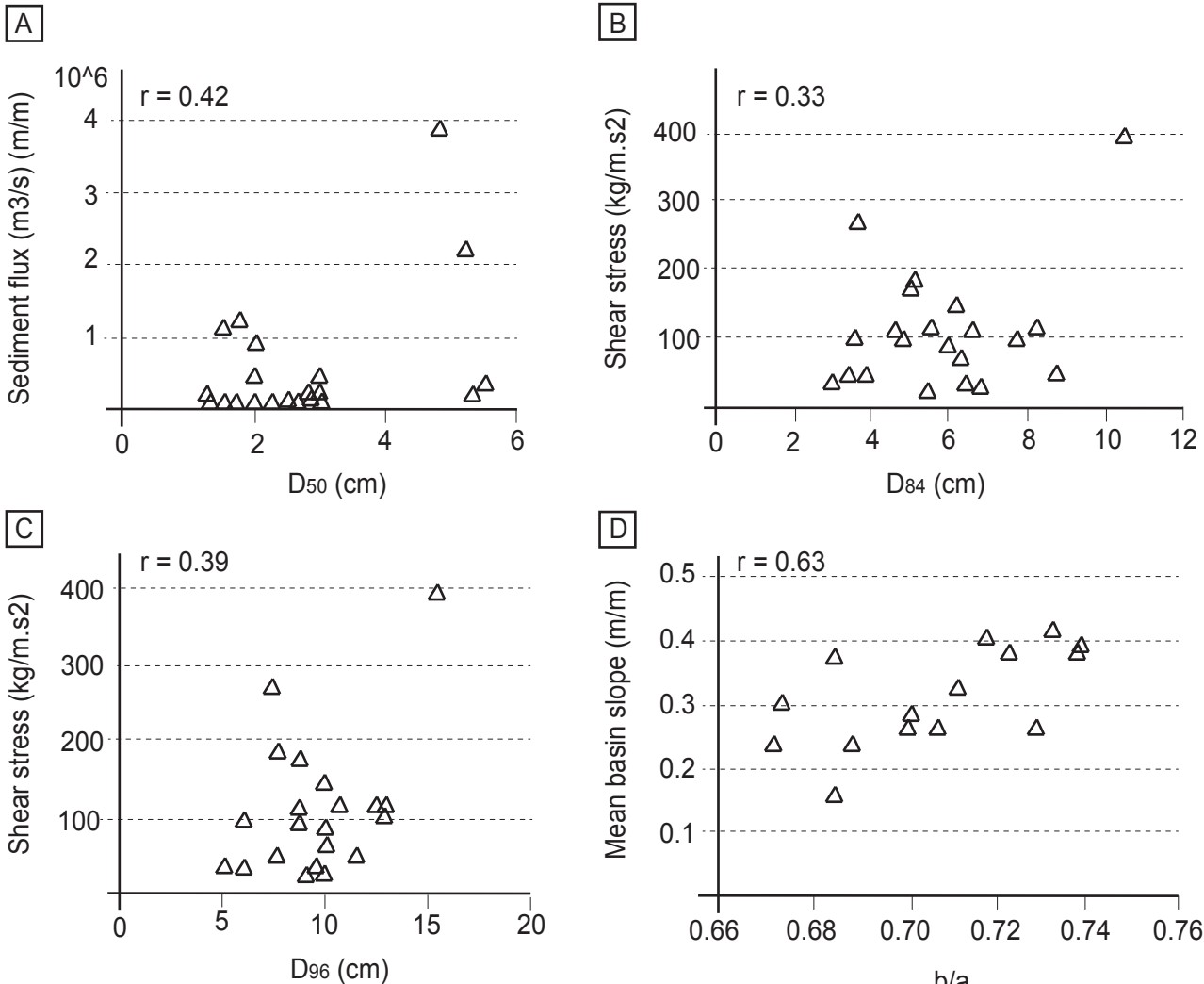

Figure 5: Correlations between the grain size data and the river parameters.
A: D50 versus sediment fluxes. B: D84 versus shear stress exerted by the water.
C: D96 versus shear stress exerted by the water.  D: Ratio b/a versus mean catchment slope.

| River name | Sample name | Altitude (m) | Latitude (DD WGS84) | Longitude (DD WGS84) | D50 (cm) | D84 (cm) | D96 (cm) | b/a | Catchment area (km2) | Mean slope (m/m) (Reber et al., in review) | Slope at the sampling site (m/m) | Distance form the western escarpment (km) | Channel width at the sampling site (m) | Mean annual water discharge (m3/s) (Reber et al., in review) | Shear stress (kg/m.s2) | Sediment flux (m3/s) | Denudation rates (mm/ka) (Reber et al., in review) | Denudation rates uncertainties (mm/ka) (Reber et al., in review) | Denudation rates corrected for Qz content in bedrock (mm/ka) (Reber et al., in review) |
|---|---|---|---|---|---|---|---|---|---|---|---|---|---|---|---|---|---|---|---|
| Tacna | PRC-ME1 | 231 | -18.12 | -70.33 | 2.3 | 6.2 | 10.0 | 0.70 | 899 | 0.28 | 0.015 | 48 | 6 | 3.4 | 142.68 | 11952 | 13.3 | 3.6 | 12.2 |
| Rio Sama Grande | PRC-ME3 | 455 | -17.82 | -70.51 | 2.5 | 5.5 | 10.6 | 0.67 | 2150 | 0.3 | 0.013 | 73 | 6 | 4 | 114.14 | 61495 | 28.6 | 5.3 | 27.7 |
| Ilo / Rio Osmore | PRC-ME5 | 1072 | -17.29 | -70.99 | 2.6 | 5.1 | 7.8 | 0.70 | 1783 | 0.26 | 0.018 | 53 | 7 | 3.4 | 184.18 | 38146 | 21.4 | 4.8 | 18.6 |
| Rio Tambo | PRC-ME6 | 145 | -17.03 | -71.69 | 1.5 | 3.6 | 7.5 | 0.69 | 12885 | 0.24 | 0.051 | 141 | 26 | 38.1 | 265.69 | 1155744 | 89.7 | 16.7 | 72.1 |
| Tambillo / Rio Sihuas | PRC-ME802 | 117 | -16.34 | -72.13 | 2.0 | 6.0 | 10.0 | 0.69 | 1708 | 0.15 | 0.019 | 70 | 15 | 30.1 | 88.78 | 58087 | 34 | 6.4 | 27.7 |
| Camana / Rio Majes | PRC-ME7 | 69 | -16.51 | -72.64 | 5.2 | 8.7 | 11.6 | 0.67 | 17401 | 0.23 | 0.005 | 188 | 100 | 68.4 | 46.06 | 2218568 | 127.5 | 23.4 | 106.8 |
| Ocona / Rio Ocona | PRC-ME9 | 14 | -16.42 | -73.12 | 4.8 | 6.8 | 10.0 | 0.71 | 16084 | 0.26 | 0.004 | 192 | 70 | 91.1 | 26.25 | 3893878 | 242.1 | 45 | 184.1 |
| Nasca / Rio Grande | PRC-ME1402 | 15 | -15.85 | -74.26 | 1.3 | 3.0 | 6.0 | 0.71 | 1412 | 0.32 | 0.014 | 48 | 3 | 20.4 | 34.10 | 65093 | 46.1 | 8.6 | 29.4 |
| Chacaltana / Rio Ica | PRC-ME15 | 3 | -15.63 | -74.64 | 2.9 | 6.4 | 9.6 | 0.73 | 4677 | 0.26 | 0.003 | 88 | 23 | 12.1 | 33.01 | 126266 | 27 | 5.7 | 25.1 |
| Humay District / Rio Pisco | PRC-ME16 | 400 | -13.73 | -75.89 | 3 | 6.6 | 13 | | 3649 | 0.34 | 0.013 | 62 | 20 | 13.6 | 112.91 | 379865 | 104.1 | 20.4 | 69.1 |
| Chinca Alta / Rio San Juan | PRC-ME17 | 75 | -13.47 | -76.14 | 1.3 | 3.8 | 7.6 | 0.69 | 3090 | 0.37 | 0.01 | 78 | 5 | 10.1 | 48.54 | 189112 | 61.2 | 11.7 | 44.1 |
| Rio Canete | PRC-ME19 | 23 | -13.12 | -76.39 | 2 | 4.6 | 8.8 | 0.72 | 6029 | 0.4 | 0.01 | 100 | 60 | 26.4 | 112.24 | 402743 | 66.8 | 12.3 | 51.2 |
| Rio Omas | PRC-ME20 | 33 | -12.67 | -76.65 | 1.6 | 4.8 | 8.8 | 0.73 | 2322 | 0.41 | 0.0076 | 78 | 22 | 8.2 | 95.14 | 62913 | 27.1 | 5.4 | 17.9 |
| Rio Lurin | PRC-ME22 | 40 | -12.25 | -76.89 | 3 | 5 | 8.8 | 0.74 | 1572 | 0.38 | 0.022 | 70 | 5 | 3.7 | 176.26 | 60515 | 38.5 | 7.1 | 23.6 |
| Lima / Rio Chillon | PRC-ME39 | 402 | -11.79 | -76.99 | 5.3 | 10.5 | 15.5 | | 1755 | 0.39 | 0.018 | 51 | 40 | 4.9 | 392.89 | 144272 | 82.2 | 15.5 | 53.4 |
| Rio Chancay | PRC-ME23 | 72 | -11.61 | -77.24 | 5.5 | 8.3 | 12.5 | 0.74 | 3059 | 0.39 | 0.01 | 66 | 20 | 8.9 | 111.55 | 298866 | 97.7 | 18.4 | 52.8 |
| Rio Supe | PRC-ME25 | 74 | -11.07 | -77.59 | 2.8 | 7.7 | 13 | | 4306 | 0.38 | 0.012 | 82 | 5 | 3.8 | 98.55 | 179550 | 41.7 | 7.7 | 25.6 |
| Rio Pativilca | PAT-ME | 10 | -10.72 | -77.77 | 1.8 | 3.6 | 6 | | 4607 | 0.44 | 0.014 | 74 | 30 | 30.9 | 96.30 | 1198281 | 260.1 | 48.8 | 190.9 |
| Huarmey | PRC-ME38 | 24 | -10.07 | -78.16 | 1.7 | 3.4 | 5.2 | | 2072 | 0.37 | 0.004 | 78 | 15 | 9.8 | 38.34 | 40816 | 19.7 | 4.5 | 10.1 |
| Rio Santa | PRC-ME27 | 80 | -8.97 | -78.62 | 2 | 5.4 | 9 | 0.72 | 12313 | 0.38 | 0.005 | 65 | 40 | 96.1 | 23.08 | 876699 | 71.2 | 13.4 | 70.4 |
| San Martin de Porres | PRC-ME30 | 67 | -7.32 | -79.48 | 2.9 | 6.3 | 10 | | 3882 | 0.34 | 0.007 | 126 | 40 | 25.4 | 65.72 | 118401 | 30.5 | 5.9 | 25.8 |

Table 1 : Location of the sampling sites with the altitude in meters above sea level.
The table also displays grain size results together with the rivers' and basins' properties and hydrological properties.
Morphometric dataset for the sampled drainage basins. All calculations are based on the 90 m resolution DEM (NASA)
The precipitation, water discharge data and the denudation rates are from Reber et al., in review

| | Distance from the coast (km) | Altitude (m) | Latitude (°) | Longitude (°) | D50 | D84 | D96 | b/a |
|---|---|---|---|---|---|---|---|---|
| GS1 | 20 | 69 | -16.51 | -72.64 | 5.2 | 8.7 | 11.6 | 0.67 |
| GS2 | 45 | 283 | -16.37 | -72.49 | 4.8 | 10 | 15 | 0.69 |
| GS3 | 57 | 378 | -16.28 | -72.45 | 5.4 | 12.7 | 21 | 0.65 |
| GS4 | 90 | 700 | -16.00 | -72.48 | 3.3 | 12 | 22.5 | 0.67 |
| GS5 | 106 | 882 | -15.86 | -72.45 | 6.2 | 19 | 31 | 0.71 |

Table 2: Location of the sampling sites in the Majes basin and grain size results in the Majes basin.

| | Altitude (m) | Latitude (DD WGS84) | Longitude (DD WGS84) | D50 (cm) | D84 (cm) | D96 (cm) | b/a | Catchment area (km2) | Mean slope (m/m) | Distance form the western escarpment (km) | Mean annual water discharge (m3/s) | Shear stress (kg/m.s2) | Sediment flux (m3/s) | Denudation rates (mm/ka) | Denudation rates corrected for Qz content in bedrock (mm/ka) |
|---|---|---|---|---|---|---|---|---|---|---|---|---|---|---|---|
| Altitude (m) | 1.00 | | | | | | | | | | | | | | |
| Latitude (DD WGS84) | -0.36 | 1.00 | | | | | | | | | | | | | |
| Longitude (DD WGS84) | 0.46 | -0.97 | 1.00 | | | | | | | | | | | | |
| D50 (cm) | 0.09 | 0.00 | -0.01 | 1.00 | | | | | | | | | | | |
| D84 (cm) | 0.14 | 0.04 | -0.03 | **0.87** | 1.00 | | | | | | | | | | |
| D96 (cm) | 0.18 | 0.02 | -0.02 | **0.73** | **0.93** | 1.00 | | | | | | | | | |
| b/a | -0.30 | 0.66 | -0.71 | 0.09 | 0.00 | -0.02 | 1.00 | | | | | | | | |
| Catchment area (km2) | -0.25 | -0.12 | 0.12 | **0.31** | 0.16 | 0.04 | -0.25 | 1.00 | | | | | | | |
| Mean slope (m/m) | -0.23 | 0.72 | -0.78 | -0.07 | -0.10 | -0.03 | **0.63** | -0.28 | 1.00 | | | | | | |
| Distance form the western escarpment (km) | -0.32 | -0.14 | 0.14 | **0.35** | 0.16 | 0.03 | **-0.33** | 0.84 | -0.35 | 1.00 | | | | | |
| Mean annual water discharge (m3/s) (Reber et al., in review) | -0.30 | 0.03 | -0.01 | 0.18 | 0.05 | -0.07 | -0.13 | 0.87 | -0.23 | 0.64 | 1.00 | | | | |
| Shear stress (kg/m.s2) | 0.45 | -0.11 | 0.14 | **0.23** | **0.33** | **0.39** | -0.06 | -0.21 | 0.06 | -0.23 | -0.37 | 1.00 | | | |
| Sediment flux (m3/s9 | -0.23 | -0.19 | 0.17 | **0.42** | 0.17 | 0.03 | -0.21 | 0.86 | -0.24 | 0.82 | 0.80 | -0.22 | 1.00 | | |
| Denudation rates (mm/ka) (Reber et al., in review) | -0.23 | 0.04 | -0.09 | **0.34** | 0.09 | 0.00 | -0.09 | 0.56 | 0.12 | 0.48 | 0.56 | -0.07 | 0.79 | 1.00 | |
| Denudation rates corrected for Qz content in bedrock (mm/ka) (Reber et al., in review) | -0.22 | 0.01 | -0.04 | 0.30 | 0.06 | -0.03 | -0.17 | 0.64 | 0.05 | 0.54 | 0.65 | -0.11 | 0.84 | 0.99 | 1.00 |

Table 3: Results of the statistical investigations, illustrated here as correlation matrix values.
The valuess in bold show significant correlation between the grain size data and the morphometric parameters and basins characteristics

| | Altitude (m) | Latitude (DD WGS84) | Longitude (DD WGS84) | D50 (cm) | D84 (cm) | D96 (cm) | b/a | Catchment area (km2) | Mean slope (m/m) | Distance form the western escarpment (km) | Mean annual water discharge (m3/s) | Shear stress | Sediment flux | Denudation rates (mm/ka) | Denudation rates corrected for Qz content in bedrock (mm/ka) |
|---|---|---|---|---|---|---|---|---|---|---|---|---|---|---|---|
| Altitude (m) | **< 0.00001** | | | | | | | | | | | | | | |
| Latitude (DD WGS84) | 0.11 | **< 0.00001** | | | | | | | | | | | | | |
| Longitude (DD WGS84) | **0.03** | **< 0.00001** | **< 0.00001** | | | | | | | | | | | | |
| D50 (cm) | 0.69 | 1.00 | 0.96 | **< 0.00001** | | | | | | | | | | | |
| D84 (cm) | 0.54 | 0.86 | 0.89 | **< 0.00001** | **< 0.00001** | | | | | | | | | | |
| D96 (cm) | 0.43 | 0.93 | 0.93 | **0.000172** | **< 0.00001** | **< 0.00001** | | | | | | | | | |
| b/a | 0.27 | **0.007** | **0.003** | 0.75 | 1 | 0.94 | **< 0.00001** | | | | | | | | |
| Catchment area (km2) | 0.27 | 0.60 | 0.60 | 0.17 | 0.48 | 0.86 | 0.37 | **< 0.00001** | | | | | | | |
| Mean slope (m/m) | 0.31 | **0.0002** | **< 0.00001** | 0.76 | 0.66 | 0.89 | **0.01** | 0.22 | **< 0.00001** | | | | | | |
| Distance form the western escarpment (km) | 0.15 | 0.54 | 0.54 | 0.11 | 0.48 | 0.89 | 0.22 | **< 0.00001** | 0.11 | **< 0.00001** | | | | | |
| Mean annual water discharge (m3/s) (Reber et al., in press) | 0.18 | 0.89 | 0.96 | 0.43 | 0.82 | 0.77 | 0.64 | **< 0.00001** | 0.31 | **< 0.00001** | **< 0.00001** | | | | |
| Shear stress | **0.04** | 0.63 | 0.54 | 0.31 | 0.14 | **0.08** | 0.83 | 0.36 | 0.79 | 0.31 | **0.098** | **< 0.00001** | | | |
| Sediment flux | 0.31 | 0.40 | 0.46 | **0.05** | 0.46 | 0.89 | 0.45 | **< 0.00001** | 0.29 | **< 0.00001** | **< 0.00001** | 0.33 | **< 0.00001** | | |
| Denudation rates (mm/ka) (Reber et al., in press) | 0.31 | 0.86 | 0.69 | 0.13 | 0.69 | 1.00 | 0.75 | **0.01** | 0.60 | **0.027** | **0.008** | 0.76 | **< 0.00001** | **< 0.00001** | |
| Denudation rates corrected for Qz content in bedrock (mm/ka) (Reber et al., in press) | 0.33 | 0.96 | 0.86 | 0.18 | 0.79 | 0.89 | 0.55 | **0.001** | 0.82 | **0.011** | **0.0014** | 0.63 | **< 0.00001** | **< 0.00001** | **< 0.00001** |

Table 4: Results of the statistical investigations, illustrated here as correlation matrix of the p-values.
The values in bold have a significance level alpha < 0.1