# Peer review of "Possible threshold controls on sediment grain properties of"

_Earth Surface Dynamics, 2017_

## Referee Comment (RC1) · Anonymous Referee #1 · 20 Mar 2017

This paper analyses variations in fluvial grain size along the western Peruvian margin. The overall idea behind the paper is interesting, and there are many significant research questions as to what controls spatial distributions of grain size through fluvial systems. However, I think that there are two main ways in which this paper needs to be improved. The first is a clearer presentation of the multiple interacting processes that explain the expected changes in grain size. The second is a more robust analysis of the data and more consideration of its limitations.

One of the main issues that I have with this paper is that it lacks a clear explanation of how the different factors that are meant to influence grain size operate, both in the introduction and throughout the discussion. For example, it is stated that increased uplift will be expected the increase grain size, but the causal mechanism is not described. Other explanations are sometimes vague. There is also a difficulty in separating out the

different mechanisms; for example, smaller basins seem to be correlated with lesser uplift, hence it is not obvious which of these two factors is more important. Another issue is that the paper seems to alternate between assuming that downstream fining is caused by abrasion and that it is caused by selective transport, without any explicit consideration of which process is likely to be more important, or the implications of one process being dominant. (A relevant paper for the discussion of abrasion processes is Sklar et al., 2006.) Overall, I would have liked a greater sense of the underlying processes that control grain size, how they interact with each other, and the relative importance of the different factors.

I also have some queries about the way in which the data were collected and analysed. The authors do not state how the locations in the different river basins were selected (other than the presence of the highway). My concern is that they are attempting to compare grains sizes that are collected from different relative locations within the basin, and are therefore not comparing like with like. For example, if the basins all had the same rate of downstream fining but the samples were collected from different locations within the basins, then the analysis would show differences between the basins that are not actually there. The authors need to consider this as a possible source of variation within their results. It would be useful to consider sample location as a function of total basin length, and also to normalise the distance to the knickpoint. There is also the question as to whether these basins are in a form of equilibrium or whether the grain size might actually reflect transient processes such as a coarse sediment slug progressing through the basin. I think that you need more discussion of the literature on controls on downstream grain size; at present the relevant papers are only referred to in passing at the start of the introduction.

Some other aspects of the analysis could also be clarified: what were the channel morphologies, how representative are the selected bars, how large were the individual images, how were grains selected within the images, how were grain outlines identified (automated or manual analysis), was any attempt made to verify the grain size data

produced, why were 500 extra grains used for grain shape, and what are the error bars on D50/D84/D95 (and hence are the identified differences significant)? The lack of a clear hypothesis early on means that some of the analysis comes across as a bit of a fishing expedition, with lots of correlations on different data groupings being undertaken, and only the significant ones being presented. I think that you need to be more thorough about this analysis, for example through multiple or stepwise regression.

Comments by line: 10: Overall the abstract could be more specific and provide some more evidence for the various claims.

53: To what extent are these different factors interrelated?

55: Make it clearer how this information about the general setting is related to the overall aim of understanding grain size.

78: Be more explicit about why uplift produces larger clasts.

79: You describe both N-S and E-W variations; which are most important for your study?

97: I'm surprised that erosion is nearly zero (line 89) given this high precipitation.

122: Be more specific about uplift rates.

125: Is five sites enough to identify trends?

196: Calculate sorting parameters to quantify these trends.

176: Suggests that you are downstream of the gravel-sand transition? Does the transition occur in other basins?

195: It would be useful to calculate stream power, as this would enable you to look at the combined impact of slope, width and discharge.

201: Is the relationship significant?

202: Was this analysis done for the other basins too?

[Figure]

209: Overall there are many competing ideas in the discussion, and it's not clear which are most important.

213: Use a sorting parameter.

216: This is the first mention of sediment sources; this needs to go earlier in the paper.

225: Note that rivers can also adjust to changes in uplift by changing other factors such as width, morphology and the amount of sediment cover.

244: What is the mechanism that relates different flood characteristics to different grain sizes?

257: What is your evidence?

273: How does the size of this fracture network compare to the grain sizes?

287: This argument would be stronger if you presented the lithological characteristics of your grains, which you could identify from the photos. Or state that they are all identical within each basin.

288: Note that you only have information on 2D grain shape not 3D.

296: Which idea do you think is more correct?

300: Is this consistent with the geological variations?

321: I'm still not entirely clear what you mean by a 'geomorphic' control.

323: But much of the earlier discussion has referred to abrasion.

Table 1: Add an indication of where the site is relative to the knickpoint and within the basin. It would help to also present distances normalise by total basin length.

Table 3: Give sorting values.

Figure 1: Add basin outlines to maps B and C.

Figure 2: Add the channel.

Sklar, L.S., Dietrich, W.E., Foufoula-Georgiou, E., Lashermes, B., Bellugi, D., 2006. Do gravel bed river size distributions record channel network structure? Water Resour. Res. 42. doi:10.1029/2006WR005035

---

## Referee Comment (RC2) · Anonymous Referee #2 · 20 Mar 2017

In this paper, the authors present a large sedimentological dataset from rivers draining the western Peruvian Andes, and attempt to find relationships between grain size/sphericity, and variables like latitude, climate, tectonic setting, runoff and catchment properties. The field data have been collected carefully, from well-chosen sites spanning an interesting part of the Andes, and I am sure there are interesting insights to be gained by studying this dataset. Unfortunately, the results, interpretations and conclusions of this paper are confusing and unconvincing for several reasons.

The authors rule out a tectonic control by simply stating that greater surface uplift rates should result in larger clast sizes. Why? The mechanism underpinning this assumption (e.g., enhanced landsliding as a result of incision, etc) is very important if you want to look for tectonic signals in sedimentological data.

The Methods section requires more information about where the grain size were collected. 'Along a highway' isn't very helpful – were the measurements made at equivalent locations in the longitudinal profiles of the catchments? If you want to compare measurements from one catchment to another, it's important to demonstrate that the data come from comparable sampling sites. It would also be helpful to know where the discharge data were collected in the catchments. I appreciate that the coordinates are listed in Table 1, but some description is needed about whether the discharge data represent equivalent points in the catchments; i.e., if one catchment is sampled at the mouth and another at the headwaters, how can a meaningful comparison be made?

I have some major criticisms of the results. Uncertainties are needed on the grain size percentiles, because the scatter in Fig. 3a is larger than the trends the authors interpret. The way the authors describe the grain size data from line 162 onwards implies a systematic variation from north to south, which is not really true. It should be clarified that the rates of grain size change from north to south refer to an average regression fitted to the data. The whole paragraph from line 171 is not really a description of results, and could be moved to the Discussion. However the final point (line 176) is very important and needs some explanation. Why are there catchments in the middle of the study area that apparently have much bigger grain size differences (only sand and no gravel) than the catchments examined in the paper? The authors are apparently aware of much larger grain size variability in the area but have ignored those catchments, and it is not clear to me why.

There are some issues with Fig. 3. The data in panel A are compressed to the bottom of the graph and half the plot isn't used – please expand the data so the reader can better see the trends (the annotations can go above the graph). In panel B, I am concerned that some of the data points are missing between 5-15 degrees latitude. Why are there only 6 points (compared to 11 in A)? Also, which percentile has been used to calculate the a/b ratio?

Next, it appears the coarsest grain sizes from the northern group of catchments are

being exported from the shorter catchments that only drain west of the western escarpment. Those with larger upstream reaches crossing the western escarpment have equivalent grain sizes to the southern catchments. This difference is quite apparent by comparing Figs 1 and 3, and may invalidate the north/south grouping of catchments.

The final part of the results contrasts Figs 5 and 6. The authors suggest that there are no correlations between grain size and the chosen parameters in Fig. 5, but that there are correlations when the catchments are grouped (Fig. 6). This isn't really a comparison, because the two figures are showing different things. I cannot tell how Fig. 5e and 5f would compare to Fig. 6 if the same normalisation was performed on discharge. Why was discharge normalised in Fig. 6a but not elsewhere in the paper? And why have the authors chosen those particular grain size percentiles and variables in Fig. 6? It seems they have simply plotted everything against everything else and shown two unrelated correlations that are not particularly convincing and do not test a particular hypothesis. I am confused about why the southern catchments should be characterised by comparing runoff normalised by area with D50, while the northern catchments should be characterised by their gradients as a function of D96.

The Discussion attempts to address some important questions about grain size patterns observed in river networks and how they might record various forcings. Unfortunately, it is inconclusive and unclear. The authors claim around line 219 that fluvial transport dominates the Majes basin – if so, why does the D50 not fine over a 100 km distance? In section 4.2, do the arguments here require that smaller rivers in smaller basins are moving coarser material? This needs to be clarified. For section 4.3, what is the actual difference in climate between the northern and southern domains? In Fig. 1c, apart from the wetter patch near Huaraz (which actually overlies a catchment exporting finer grain sizes!), the two areas look similar. I recommend the authors plot the runoff data and/or precipitation against latitude (following Fig. 3) if they want to argue there is a relationship here. They need to show that the two domains are actually different and that climate correlates with grain size if they want to make that argument. In section 4.4, the authors could clarify whether the smaller catchments in the northern group were glaciated as well, or only the larger ones? Because the coarsest data seems to only come from the smaller catchments, and this is an important difference that needs to be addressed. These smaller catchments also drain proportionately more of the Coastal Batholith, which might indicate an erodibility control on grain size. The arguments in this section are vague and undeveloped and jump from glaciers to lithology without offering any precise interpretations.

In summary, this paper presents an interesting set of field data from an interesting study area in the Andes where many good questions could be tested. However it does not develop a clear hypothesis, clear interpretations, or clear conclusions. The reader is left with an impression that most of the data show no correlations, but if you plot every variable against every other variable and divide up the dataset enough, eventually you will find some weak trends. We learn that particle sizes and sphericity might not be related to tectonics, and might be related to climate, or catchment size, or the internal dynamics of catchments and sediment supply processes, or could be related to lithology, or glaciers, or El Niño. The paper needs much clearer answers to more precise questions, but I do encourage the authors to take a fresh look at the data.

I have some minor comments that the authors might find helpful for future submissions:

- "Contrariwise" is an unusual word, and I recommend using something like "on the contrary" instead

- Refer to "El Niño", not "the El Niño" or "the El Niño effect" (it is not an effect). Also, on line 114 you equate El Niño with ENSO – they are not exactly the same thing. El Niño is one phase of ENSO and brings particular weather patterns, but ENSO refers to the overall oscillation between El Niño, neutral, and La Niña states in the tropical Pacific

- "Strong precipitation rate" implies a high intensity of precipitation, which is quite different to a greater overall amount of precipitation

**ESurfD**
- Lines 107-109. This is confusing – hot air cannot rise and is trapped against the foothills, but also cools at high altitude?

- Line 112. If you refer to Pisco, mark it on the map

- Line 143. The D96 is not the maximum particle size

- Line 183. This sentence makes a big claim and needs to be supported by some key citations

- Line 293. Is the fracture spacing 10-20cm? Because this is the particle size range. I'm sure fracture spacing sets the sizes of large boulders, but I'm not convinced this mechanism applies to pebbles

- Line 295. The authors state that abrasion makes particles more spherical, and then say it doesn't. Please clarify which it is

- Line 300. Yet the southernmost catchments in the southern grouping are very small, but show the roundest clasts. Is this not contradictory?

- Fig. 5. These axes should be reversed

---

## Author Comment (AC1) · 4 May 2017

Dear referee,

Thank you very much for handling our paper. We considered the comments as very constructive and have improved the paper accordingly. The major changes include the improvement of the introduction with a clear explanation of how tectonics and climate operate to potentially influence the grain size pattern. Based on this, we phrased a distinct hypothesis to be tested. We also we improved the methods part by adding additional information about the sampling strategy and the data collection. We have used the Pearson's correlation coefficient to obtain statistically robust correlation between our grain size data and the morphological characteristics of the basins including mean basin slope, denudation rate and basin size, and shear stresses exerted by

the streams. We found distinct correlations between the grain size pattern and these variables and have framed the discussion accordingly. Therefore, we found the grouping of basins into northern and southern domains no longer as useful and have thus re-structured the paper accordingly. In summary, the major changes include: ć Presentation of a clear outline of how tectonics and climate could influence the grain size pattern, and based on this, a formulation of a distinct hypthothesis ć Presentation of more details of how we have collected and analysed the data ć Testing through state-of-the art statistical methods whether basin shape, sediment flux and streams' shear stresses have a measurable control on the grain size pattern. We have thus re-structured the discussion part accordingy.

Please find below a point-by-point response of how we have handled the suggestions and comments. Thank you very much for your hard work. On behalf of the co-authors

Camille Litty

Response to Referee #1

It lacks a clear explanation of how the different factors that are meant to influence grain size operate, both in the introduction and throughout the discussion. For example, it is stated that increased uplift will be expected the increase grain size, but the causal mechanism is not described.

We have addressed this point by adding a new paragraph in the introduction, which explains how tectonics and earthquake occurrence should influence the grain size pattern, and what we expect based on this. In the same sense, we have discussed how this should imprint the grain size pattern. Based on this, we were able to phrase distinct hypotheses to be tested.

There is also a difficulty in separating out the different mechanisms; for example, smaller basins seem to be correlated with lesser uplift, hence it is not obvious which of these two factors is more important.

This has been confusing, indeed. We thus have completely modified the analysis plus we have framed the discussion in a different way.

Another issue is that the paper seems to alternate between assuming that downstream fining is caused by abrasion and that it is caused by selective transport, without any explicit consideration of which process is likely to be more important, or the implications of one process being dominant. (A relevant paper for the discussion of abrasion processes is Sklar et al., 2006.)

It is true that we did not take into consideration the different processes. We have clarified this point and have been consistent in our interpretation.

Overall, I would have liked a greater sense of the underlying processes that control grain size, how they interact with each other, and the relative importance of the different factors.

This has been done. We have rephrased the discussion section, thereby addressing the interplay between the controls of the various variables more carefully.

I also have some queries about the way in which the data were collected and analysed. The authors do not state how the locations in the different river basins were selected (other than the presence of the highway).

This information has been added. In fact, we have sampled all streams where upstream basin sizes were larger than 700 km2, and we have focussed our data collection at the downstream end where these rivers cross the tip of the mountain belt. This strategy allows us to explore how the ensemble of all processes in a basin relevant for the supply of material influences the grain size patter. This has been clarified in the revised version of the paper.

My concern is that they are attempting to compare grains sizes that are collected from different relative locations within the basin, and are therefore not comparing like with like. For example, if the basins all had the same rate of downstream fining but the

samples were collected from different locations within the basins, then the analysis would show differences between the basins that are not actually there. The authors need to consider this as a possible source of variation within their results.

This could indeed add a bias, however, we have selected streams where they cross the tip of the Andean mountain belt. Please see also comment above.

It would be useful to consider sample location as a function of total basin length, and also to normalise the distance to the knickpoint.

We have considered this variable (distance from edge of Western Escarpment). Please see revised version. We have not performed this normalization, but used other variables instead (e.g., shear stresses, basin-averaged denudation rates), which yield measures for flow strengths and sediment flux. Because grain size and fining trends potentially depend on these variables, we used these variables for our analysis and we have indeed found positive correlations with grain size patterns.

There is also the question as to whether these basins are in a form of equilibrium or whether the grain size might actually reflect transient processes such as a coarse sediment slug progressing through the basin. I think that you need more discussion of the literature on controls on downstream grain size; at present the relevant papers are only referred to in passing at the start of the introduction.

This might work for individual basins, such as exemplified for Majes, where the grain size decreases downstream. However, this does not work if all basins along the western Peruvian margin of the Andes are considered, because the D50, as an example, increases with downstream distance from the uppermost edge of the Western Escarpment. In fact, we would have expected the opposite where grain sizes decrease with increasing transport distance. However, we found positive correlations with grain size and mean basin slope, mean basin denudation rates and shear stresses of the streams. This suggests that supply of material (higher denudation rates) and water flow strengths have a large influence on the downstream fining trends within each basin. We

have thus framed the discussion in this direction.

What were the channel morphologies

The channels have a braided pattern, and the morphology of the longitudinal stream profiles is characterized by two segments separated by a distinct knickzone. Please see revised version.

how large were the individual images ?

This has been clarified: Individual images are about 1 m2.

how were grains selected within the images ?

Every pebble, which was entirely visible on the digital images, has been measured.

how representative are the selected bars ?

For these basins, sampling sites were situated in the trunk streams of these valleys where the streams cross the tip of the mountain belt, which is located near the Pacific Coast in most cases. We selected the downstream end of these streams because the grain size pattern at these sites is likely to record the ensemble of the main conditions and forces controlling the supply of material to the trunk stream in the upstream basin and thus the grain size caliber of these streams where they leave the Andes. In these streams, we randomly selected c. 5 longitudinal bars where we collected our grain size dataset. As such, we consider the selected bars as representative for the ensemble of supply and transport processes in the streams' basins.

how were grain outlines identified (automated or manual analysis) ?

From those photos, the intermediate b-axes and the long a-axes of around 500 pebbles were manually measured. We have added this information in the revised version.

was any attempt made to verify the grain size data produced,

No attempt has been made to verify the grain size data produced for this paper.

Nonetheless, all the pebbles have been measured by the same operator. This yields the same bias for every sampling site, if there is any.

why were 500 extra grains used for grain shape

We have clarified this in the method part

and what are the error bars on D50/D84/D95 (and hence are the identified differences significant)?

Uncertainties on the grain size percentiles are also about 3 mm. This value corresponds to the precision limits of the measurements with the software ImageJ and of the digital pictures' resolution. For the significance of the difference, correlations or trends have all been estimated through the Pearson correlation coefficient (p-value) and not anymore on a visual estimation as we have don before.

The lack of a clear hypothesis early on means that some of the analysis comes across as a bit of a fishing expedition, with lots of correlations on different data groupings being undertaken, and only the significant ones being presented. I think that you need to be more thorough about this analysis, for example through multiple or stepwise regression.

This has been done, and a hypothesis has been phrased. Please see also comment above.

Comments by line: 10: Overall the abstract could be more specific and provide some more evidence for the various claims.

We have addressed this point

53: To what extent are these different factors interrelated?

We have addressed this point by adding a new paragraph in the introduction

55: Make it clearer how this information about the general setting is related to the overall aim of understanding grain size.

Done

78: Be more explicit about why uplift produces larger clasts.

This has been specified

79: You describe both N-S and E-W variations; which are most important for your study?

N-S are more relevant; we have rephrased the introduction and clarified this point.

97: I'm surprised that erosion is nearly zero (line 89) given this high precipitation.

Abbühl et al. (2011, ESPL) have shown that the low denudation rates are due to the flat landscapes on the Altiplano.

122: Be more specific about uplift rates.

This has been changed accordingly

125: Is five sites enough to identify trends?

We have changed our data interpretation as there was no real reason to separate the basins into 2 groups (i.e. northern and southern domains). We have worked with all our dataset.

196: Calculate sorting parameters to quantify these trends.

No trend actually exists as there is no real change in the grain size from north to south, so we also did not introduce the sorting parameter.

176: Suggests that you are downstream of the gravel-sand transition? Does the transition occur in other basins?

This transition seems to not occur in the other basins. We have addressed this point in the discussion part.

195: It would be useful to calculate stream power, as this would enable you to look at

the combined impact of slope, width and discharge.

Yes, indeed, but we have calculated shear stresses instead. We have done so and we do see correlation in between the grain size and the shear stress.

201: Is the relationship significant?

Indeed, we are not considering anymore the grouping (northern and southern domains) of basins.

209: Overall there are many competing ideas in the discussion, and it's not clear which are most important.

This has been confusing, indeed. We thus have completely modified this part of the analysis plus have framed the discussion in a different way.

216: This is the first mention of sediment sources; this needs to go earlier in the paper.

We now mention it earlier in the text. 'The upstream edges of this knickzone called the Western Escarpment also delineate the upper boundaries of the major sediment sources'

225: Note that rivers can also adjust to changes in uplift by changing other factors such as width, morphology and the amount of sediment cover.

Yes we have changed the part on the tectonic control on grain size

244: What is the mechanism that relates different flood characteristics to different grain sizes?

We have rephrased the entire discussion and have likewise changed this section.

257: What is your evidence?

Because we have only found a correlation between the D50 and the basins scale properties (basin area, denudation rates, mean slope, we infer that the mean grain size reflects the ensemble of a complex pattern of erosional processes operating in the

Peruvian basins

273: How does the size of this fracture network compare to the grain sizes?

We had no indication of the size of the fabric network. We have removed this part of the discussion as we found more compelling evidence for correlations with other variables.

287: This argument would be stronger if you presented the lithological characteristics of your grains, which you could identify from the photos. Or state that they are all identical within each basin. 300: Is this consistent with the geological variations?

A test of the inferred positive correlation between mean basin slope, bedrock lithology and particularly the occurrence of plutonic rocks, and the pebbles' sphericity would require a higher resolution topographic and geologic data, which are currently not available, we thus decided to remove this part, which also does not fit anymore in the discussion, as the grouping of basins into southern and northern domains is not considered anymore.

288: Note that you only have information on 2D grain shape not 3D.

Yes indeed, these are the a- and b-axis. So we are indeed missing the information about the third dimension to talk about the shape of the clasts. In this sense, the reviewer is correct. Nevertheless, we are still convinced that the 2D info contains valuable information about the shape of the clasts in the sense that preferential abrasion due to an inherited fabric (fractions, bedding, schistosity) returns elliptical rather than spherical clasts. We have thus kept this part of our analysis.

296: Which idea do you think is more correct?

This point has been addressed in the revised version of the text.

321: I'm still not entirely clear what you mean by a 'geomorphic' control.

It was indeed unclear, we have rephrased that. But what we wanted to say is that the geomorphic parameters (basin slopes, size, denudation) were controlling the grain size

distribution

323: But much of the earlier discussion has referred to abrasion.

We have indeed been contradictory. However, we have substantially changed the paper and thus also the conclusions.

Table 1: Add an indication of where the site is relative to the knickpoint and within the basin.

This has been done in the method part

It would help to also present distances normalise by total basin length.

We did not normalize by the basin length because this is one of the parameters that we wanted to test as control on the grain size

Table 3: Give sorting values.

We have deleted table 3 as we do not group the basins into northern and southern basins

Figure 1: Add basin outlines to maps B and C.

This has been made

Figure 2: Add the channel.

This has been made

[Figure]

**Figure 1: A:** Map of the studied basins showing the sampling sites and the western escarpment (western escarpment modified after Trauerstein et al., 2013). **B:** Geological map of the western Peruvian Andes. **C:** Map of the precipitation rates showing the spatial extend of the ITCZ, modified after Huffman et al., 2007.)

**Fig. 1.**

[Figure]

**Figure 2:** Geological map of the Majes basin overlain by the precipitation pattern (Precipitation data from Steffen et al., 2010., where the black dashed lines show precipitation rates (mm/yr). GS1 to GS5 represent sites where grain size data has been collected. The right corner shows the Majes river long profile.

**Fig. 2.**

[Figure]

* Data from Reber et al., in review

Figure 3: Topography of subducting Nazca plate, where slab depth data has been extracted from earthquake.usgs.gov/data/slab/. This N-S projection also illustrates: a) tectonic lineaments such as submarine ridges and MFZ: Mendaña Fracture Zone; NFZ: Nazca Fracture Zone; b) Holocene Volcanoes; c) Earthquake data, taken from earthquake.usgs.gov/earthquakes/search/; number of earthquakes M>4 within 30 km radius window; d) Coastal elevation. The data has been extracted from a 20 km-wide swath profile along the coast. The three lines represent maximum, mean and minimum elevations within the selected swath; e) Catchment averaged denudation rates have been corrected for quartz contents; f) Mean annual precipitation rates;
g) Mean annual water discharge; h) Mean basin slope i)Grain size results for the intermediate (b)-axis of the pebbles in the streams from north to south at the sampling sites presented in Figure 1; j) Ratio between the intermediate axis and the long (a)-axis (modified after Reber et al., in review).

**Fig. 3.**

[Figure]

**Figure 4:** Grain size results along the Majes River.

**Fig. 4.**

| River name | Sample name | Altitude (m) | Latitude (DD WGS84) | Longitude (DD WGS84) | D50 (cm) | D84 (cm) | D96 (cm) | b/a | Catchment area (km2) | Mean elevation (m.a.s.l.) | Mean slope (m/m) | Slope at the sampling site (m/m) | Distance form the western escarpment (km) | Mean annual precipitation (mm/yr) (Reber et al., in review) | Mean annual water discharge (m3/s) (Reber et al., in review) | Denudation rates (mm/ka) (Reber et al., in review) | Denudation rates uncertainties (mm/ka) (Reber et al., in review) | Denudation rates corrected for Qz content in bedrock (mm/ka) (Reber et al., in review) |
|---|---|---|---|---|---|---|---|---|---|---|---|---|---|---|---|---|---|---|
| Tacna | PRC-ME1 | 231 | -18.12 | -70.33 | 2.3 | 6.2 | 10.0 | 0.70 | 899 | 2733 | 0.28 | 0.015 | 48 | 149.6 | 3.4 | 13.3 | 3.6 | 12.2 |
| Rio Sama Grande | PRC-ME3 | 455 | -17.82 | -70.51 | 2.5 | 5.5 | 10.6 | 0.67 | 2150 | 3105 | 0.3 | 0.013 | 73 | 136.4 | 4 | 28.6 | 5.3 | 27.7 |
| Ilo / Rio Osmore | PRC-ME5 | 1072 | -17.29 | -70.99 | 2.6 | 5.1 | 7.8 | 0.70 | 1783 | 3398 | 0.26 | 0.018 | 53 | 137.8 | 3.4 | 21.4 | 4.8 | 18.6 |
| Rio Tambo | PRC-ME6 | 145 | -17.03 | -71.69 | 1.5 | 3.6 | 7.5 | 0.69 | 12885 | 3568 | 0.24 | 0.051 | 141 | 216.3 | 38.1 | 89.7 | 16.7 | 72.1 |
| Tambillo / Rio Sihuas | PRC-ME802 | 117 | -16.34 | -72.13 | 2.0 | 6.0 | 10.0 | 0.69 | 1708 | 3285 | 0.15 | 0.019 | 70 | 170.2 | 30.1 | 34 | 6.4 | 27.7 |
| Camana / Rio Majes | PRC-ME7 | 69 | -16.51 | -72.64 | 5.2 | 8.7 | 11.6 | 0.67 | 17401 | 3635 | 0.23 | 0.005 | 188 | 283.9 | 68.4 | 127.5 | 23.4 | 106.8 |
| Ocona / Rio Ocona | PRC-ME9 | 14 | -16.42 | -73.12 | 4.8 | 6.8 | 10.0 | 0.71 | 16084 | 3745 | 0.26 | 0.004 | 192 | 414.7 | 91.1 | 242.1 | 45 | 184.1 |
| Nasca / Rio Grande | PRC-ME1402 | 15 | -15.85 | -74.26 | 1.3 | 3.0 | 6.0 | 0.71 | 1412 | 2716 | 0.32 | 0.014 | 48 | 283.6 | 20.4 | 46.1 | 8.6 | 29.4 |
| Chacaltana / Rio Ica | PRC-ME15 | 3 | -15.63 | -74.64 | 2.9 | 6.4 | 9.6 | 0.73 | 4677 | 2204 | 0.26 | 0.003 | 88 | 188.4 | 12.1 | 27 | 5.7 | 25.1 |
| Humay District / Rio Pisco | PRC-ME16 | 400 | -13.73 | -75.89 | 3 | 6.6 | 13 |  | 3649 | 3464 | 0.34 | 0.013 | 62 | 272.6 | 13.6 | 104.1 | 20.4 | 69.1 |
| Chinca Alta / Rio San Juan | PRC-ME17 | 75 | -13.47 | -76.14 | 1.3 | 3.8 | 7.6 | 0.69 | 3090 | 3197 | 0.37 | 0.01 | 78 | 237.8 | 10.1 | 61.2 | 11.7 | 44.1 |
| Rio Canete | PRC-ME19 | 23 | -13.12 | -76.39 | 2 | 4.6 | 8.8 | 0.72 | 6029 | 3648 | 0.4 | 0.01 | 100 | 318.4 | 26.4 | 66.8 | 12.3 | 51.2 |
| Rio Omas | PRC-ME20 | 33 | -12.67 | -76.65 | 1.6 | 4.8 | 8.8 | 0.73 | 2322 | 3294 | 0.41 | 0.0076 | 78 | 257.6 | 8.2 | 27.1 | 5.4 | 17.9 |
| Rio Lurin | PRC-ME22 | 40 | -12.25 | -76.89 | 3 | 5 | 8.8 | 0.74 | 1572 | 2568 | 0.38 | 0.022 | 70 | 175.5 | 3.7 | 38.5 | 7.1 | 23.6 |
| Lima / Rio Chillon | PRC-ME39 | 402 | -11.79 | -76.99 | 5.3 | 10.5 | 15.5 |  | 1755 | 2942 | 0.39 | 0.018 | 51 | 204.7 | 4.9 | 82.2 | 15.5 | 53.4 |
| Rio Chancay | PRC-ME23 | 72 | -11.61 | -77.24 | 5.5 | 8.3 | 12.5 | 0.74 | 3059 | 2697 | 0.39 | 0.01 | 66 | 211.4 | 8.9 | 97.7 | 18.4 | 52.8 |
| Rio Supe | PRC-ME25 | 74 | -11.07 | -77.59 | 2.8 | 7.7 | 13 |  | 4306 | 2365 | 0.38 | 0.012 | 82 | 275.4 | 3.8 | 41.7 | 7.7 | 25.6 |
| Rio Pativilca | PAT-ME | 10 | -10.72 | -77.77 | 1.8 | 3.6 | 6 |  | 4607 | 3378 | 0.44 | 0.014 | 74 | 490.6 | 30.9 | 260.1 | 48.8 | 190.9 |
| Huarmey | PRC-ME38 | 24 | -10.07 | -78.16 | 1.7 | 3.4 | 5.2 |  | 2072 | 2337 | 0.37 | 0.004 | 78 | 340.1 | 9.8 | 19.7 | 4.5 | 10.1 |
| Rio Santa | PRC-ME27 | 80 | -8.97 | -78.62 | 2 | 5.4 | 9 | 0.72 | 12313 | 3262 | 0.38 | 0.005 | 65 | 571.7 | 96.1 | 71.2 | 13.4 | 70.4 |
| San Martin de Porres | PRC-ME30 | 67 | -7.32 | -79.48 | 2.9 | 6.3 | 10 |  | 3882 | 2292 | 0.34 | 0.007 | 126 | 472.8 | 25.4 | 30.5 | 5.9 | 25.8 |

Table 1 : Location of the sampling sites with the altitude in meters above sea level.
The table also displays grain size results together with the rivers' and basins' properties and hydrological properties.
Morphometric dataset for the sampled drainage basins. All calculations are based on the 90 m resolution DEM (NASA)
The precipitation, water discharge data and the denudation rates are from Reber et al., in review

**Fig. 5.**

| | Distance from the coast (km) | Altitude (m) | Latitude (°) | Longitude (°) | D50 | D84 | D96 | b/a |
|---|---|---|---|---|---|---|---|---|
| GS1 | 20 | 69 | -16.51 | -72.64 | 5.2 | 8.7 | 11.6 | 0.67 |
| GS2 | 45 | 283 | -16.37 | -72.49 | 4.8 | 10 | 15 | 0.69 |
| GS3 | 57 | 378 | -16.28 | -72.45 | 5.4 | 12.7 | 21 | 0.65 |
| GS4 | 90 | 700 | -16.00 | -72.48 | 3.3 | 12 | 22.5 | 0.67 |
| GS5 | 106 | 882 | -15.86 | -72.45 | 6.2 | 19 | 31 | 0.71 |

Table 2: Location of the sampling sites in the Majes basin and grain size results in the Majes basin.

**Fig. 6.**

| | *Altitude (m)* | *Latitude (DD WGS84)* | *Longitude (DD WGS84)* | ***D50 (cm)*** | ***D84 (cm)*** | ***D96 (cm)*** | ***b/a*** | *Catchment area (km2)* | *Mean elevation (m.a.s.l.)* | *Mean slope (m/m)* | *Distance form the western escarpment (km)* | *Mean annual precipitation (mm/yr)* | *Mean annual water discharge (m3/s)* | *Shear stress* | *Denudation rates (mm/ka)* | *Denudation rates corrected for Qz content in bedrock (mm/ka)* |
|---|---|---|---|---|---|---|---|---|---|---|---|---|---|---|---|---|
| Altitude (m) | 1.00 | | | | | | | | | | | | | | | |
| Latitude (DD WGS84) | -0.36 | 1.00 | | | | | | | | | | | | | | |
| Longitude (DD WGS84) | 0.46 | -0.97 | 1.00 | | | | | | | | | | | | | |
| D50 (cm) | 0.09 | 0.00 | -0.01 | 1.00 | | | | | | | | | | | | |
| D84 (cm) | 0.14 | 0.04 | -0.03 | **0.87** | 1.00 | | | | | | | | | | | |
| D96 (cm) | 0.18 | 0.02 | -0.02 | **0.73** | **0.93** | 1.00 | | | | | | | | | | |
| b/a | -0.30 | 0.66 | -0.71 | 0.09 | 0.00 | -0.02 | 1.00 | | | | | | | | | |
| Catchment area (km2) | -0.25 | -0.12 | 0.12 | **0.31** | 0.16 | 0.04 | -0.25 | 1.00 | | | | | | | | |
| Mean elevation (m.a.s.l.) | 0.21 | -0.38 | 0.37 | 0.03 | -0.07 | -0.03 | **-0.49** | 0.53 | 1.00 | | | | | | | |
| Mean slope (m/m) | -0.23 | 0.72 | -0.78 | -0.07 | -0.10 | -0.03 | **0.63** | -0.28 | -0.20 | 1.00 | | | | | | |
| Distance form the western escarpment (km) | -0.32 | -0.14 | 0.14 | **0.35** | 0.16 | 0.03 | **-0.33** | 0.84 | 0.37 | -0.35 | 1.00 | | | | | |
| Mean annual precipitation (mm/yr) (Reber et al., in review) | -0.43 | 0.65 | -0.61 | -0.08 | -0.16 | **-0.23** | 0.21 | 0.44 | 0.11 | 0.39 | 0.30 | 1.00 | | | | |
| Mean annual water discharge (m3/s) (Reber et al., in review) | -0.30 | 0.03 | -0.01 | 0.18 | 0.05 | -0.07 | -0.13 | 0.87 | 0.51 | -0.23 | 0.64 | 0.66 | 1.00 | | | |
| Shear stress | 0.45 | -0.11 | 0.14 | **0.23** | **0.33** | **0.39** | -0.06 | -0.21 | 0.09 | 0.06 | -0.23 | -0.43 | -0.37 | 1.00 | | |
| Denudation rates (mm/ka) (Reber et al., in review) | -0.23 | 0.04 | -0.09 | **0.34** | 0.09 | 0.00 | -0.09 | 0.56 | 0.53 | 0.12 | 0.48 | 0.50 | 0.56 | -0.07 | 1.00 | |
| Denudation rates corrected for Qz content in bedrock (mm/ka) (Reber et al., in review) | -0.22 | 0.01 | -0.04 | 0.30 | 0.06 | -0.03 | -0.17 | 0.64 | 0.57 | 0.05 | 0.54 | 0.54 | 0.65 | -0.11 | 0.99 | 1.00 |

Table 3: Results of the statistical investigations, illustrated here as correlation matrix values.
The valuess in bold show significant correlation between the grain size data and the morphometric parameters and basins characteristics

**Fig. 7.**

---

## Author Comment (AC2) · 4 May 2017

Dear referee,

Thank you very much for handling our paper. We considered the comments as very constructive and have improved the paper accordingly. The major changes include the improvement of the introduction with a clear explanation of how tectonics and climate operate to potentially influence the grain size pattern. Based on this, we phrased a distinct hypothesis to be tested. We also we improved the methods part by adding additional information about the sampling strategy and the data collection. We have used the Pearson's correlation coefficient to obtain statistically robust correlation between our grain size data and the morphological characteristics of the basins including mean basin slope, denudation rate and basin size, and shear stresses exerted by

the streams. We found distinct correlations between the grain size pattern and these variables and have framed the discussion accordingly. Therefore, we found the grouping of basins into northern and southern domains no longer as useful and have thus re-structured the paper accordingly. In summary, the major changes include: • Presentation of a clear outline of how tectonics and climate could influence the grain size pattern, and based on this, a formulation of a distinct hypthothesis • Presentation of more details of how we have collected and analysed the data • Testing through state-of-the art statistical methods whether basin shape, sediment flux and streams' shear stresses have a measurable control on the grain size pattern. We have thus re-structured the discussion part accordingy.

Please find below a point-by-point response of how we have handled the suggestions and comments. Thank you very much for your hard work. On behalf of the co-authors

Camille Litty

Response to Referee #2

The authors rule out a tectonic control by simply stating that greater surface uplift rates should result in larger clast sizes. Why?

This has been explained in the introduction, which we have sufficiently modified thereby addressing the points of reviewer 1.

The mechanism underpinning this assumption (e.g., enhanced landsliding as a result of incision, etc) is very important if you want to look for tectonic signals in sedimentological data.

Yes indeed; we thus have modified the introduction accordingly.

The Methods section requires more information about where the grain size were collected.

Improved and expanded.

'Along a highway' isn't very helpful – were the measurements made at equivalent locations in the longitudinal profiles of the catchments?

Done and improved.

If you want to compare measurements from one catchment to another, it's important to demonstrate that the data come from comparable sampling sites.

Yes, done and we have specified this point.

It would also be helpful to know where the discharge data were collected in the catchments. I appreciate that the coordinates are listed in Table 1, but some description is needed about whether the discharge data represent equivalent points in the catchments; i.e., if one catchment is sampled at the mouth and another at the headwaters, how can a meaningful comparison be made?

We have taken the discharge data from Reber et al., in review in Terra Nova. These authors provide the full information about the data source.

I have some major criticisms of the results. Uncertainties are needed on the grain size percentiles, because the scatter in Fig. 3a is larger than the trends the authors interpret.

The grain size data have too large a scatter, so interpretations of trends are indeed not possible. We have changed the manuscript accordingly. We have worked on statistical correlations using the Pearson's coefficient and no correlation has been found between the D50 and the latitude. Uncertainties on the grain size percentiles are also about 3 mm. This value corresponds to the precision limits of the measurements with the software ImageJ and of the digital pictures resolution.

The way the authors describe the grain size data from line 162 onwards implies a systematic variation from north to south, which is not really true.

Indeed, we have changed the analyses, and there is indeed not such a trend.

It should be clarified that the rates of grain size change from north to south refer to an average regression fitted to the data.

We have changed the analyses, and there is not such a trend.

The whole paragraph from line 171 is not really a description of results, and could be moved to the Discussion.

We have changed the analyses, so this paragraph has been removed.

However the final point (line 176) is very important and needs some explanation.

We have added an all paragraph on the gravel front in the discussion

Why are there catchments in the middle of the study area that apparently have much bigger grain size differences (only sand and no gravel) than the catchments examined in the paper?

We have discussed this point.

The authors are apparently aware of much larger grain size variability in the area but have ignored those catchments, and it is not clear to me why.

We have discussed this point.

There are some issues with Fig. 3. The data in panel A are compressed to the bottom of the graph and half the plot isn't used – please expand the data so the reader can better see the trends (the annotations can go above the graph). In panel B, I am concerned that some of the data points are missing between 5-15 degrees latitude. Why are there only 6 points (compared to 11 in A)?

Figure 3 has been improved. The ratio b/a has not been measured at each sampling site so there are less data points in the ratio plot than in the percentiles one.

Also, which percentile has been used to calculate the a/b ratio?

There is no percentile used. We measured the length of the a-axis and the b-axis were

per pebble. This gives us one value for the ratio. We repeated this for 500 pebbles, yielding a mean value per sampling site.

Next, it appears the coarsest grain sizes from the northern group of catchments are being exported from the shorter catchments that only drain west of the western escarpment. Those with larger upstream reaches crossing the western escarpment have equivalent grain sizes to the southern catchments. This difference is quite apparent by comparing Figs 1 and 3, and may invalidate the north/south grouping of catchments.

Yes indeed; we also realized that and have rewritten the discussion part of the paper.

The final part of the results contrasts Figs 5 and 6.

Yes indeed. We have completely changed this part of the analysis

The authors suggest that there are no correlations between grain size and the chosen parameters in

Fig. 5, but that there are correlations when the catchments are grouped (Fig. 6).

We have changed this part of the analysis

This isn't really a comparison, because the two figures are showing different things. I cannot tell how Fig. 5e and 5f would compare to Fig. 6 if the same normalisation was performed on discharge.

Indeed, please note that we have changed this part of the analysis

Why was discharge normalised in Fig. 6a but not elsewhere in the paper? And why have the authors chosen those particular grain size percentiles and variables in Fig. 6?

This has been changed and corrected. Indeed, this did not make sense.

It seems they have simply plotted everything against everything else and shown two unrelated correlations that are not particularly convincing and do not test a particular hypothesis.

We have framed our paper around a hypothesis. So this aspect has been changed. We have made a new figure showing the data from south to north and a correlation matrix using the Pearson's correlation coefficient to give statistically robust analyses.

I am confused about why the southern catchments should be characterised by comparing runoff normalised by area with D50, while the northern catchments should be characterised by their gradients as a function of D96.

We have changed this part of the analysis

The Discussion attempts to address some important questions about grain size patterns observed in river networks and how they might record various forcings. Unfortunately, it is inconclusive and unclear. The authors claim around line 219 that fluvial transport dominates the Majes basin – if so, why does the D50 not fine over a 100 km distance?

We have addressed this point.

In section 4.2, do the arguments here require that smaller rivers in smaller basins are moving coarser material? This needs to be clarified.

We have changed this part of the analysis

For section 4.3, what is the actual difference in climate between the northern and southern domains?

We have changed this part of the analysis; we no longer perform this grouping.

In Fig. 1c, apart from the wetter patch near Huaraz (which actually overlies a catchment exporting finer grain sizes!), the two areas look similar. I recommend the authors plot the runoff data and/or precipitation against latitude (following Fig. 3) if they want to argue there is a relationship here.

We have changed this part of the analysis

They need to show that the two domains are actually different and that climate correlates with grain size if they want to make that argument.

We changed this discussion accordingly

In section 4.4, the authors could clarify whether the smaller catchments in the northern group were glaciated as well, or only the larger ones? Because the coarsest data seems to only come from the smaller catchments, and this is an important difference that needs to be addressed.

We have clarify this point

These smaller catchments also drain proportionately more of the Coastal Batholith, which might indicate an erodibility control on grain size.

We have changed this part of the analysis and the discussion accordingly.

The arguments in this section are vague and undeveloped and jump from glaciers to lithology without offering any precise interpretations.

Yes, indeed. We have removed this part as it was non-conclusive.

- "Contrariwise" is an unusual word, and I recommend using something like "on the contrary" instead

We have learned this word from an English native speaker, so we have kept it.

- Refer to "El Niño", not "the El Niño" or "the El Niño effect" (it is not an effect). Also, on line 114 you equate El Niño with ENSO – they are not exactly the same thing. El Niño is one phase of ENSO and brings particular weather patterns, but ENSO refers to the overall oscillation between El Niño, neutral, and La Niña states in the tropical Pacific

Yes, indeed. We have removed this part.

- "Strong precipitation rate" implies a high intensity of precipitation, which is quite different to a greater overall amount of precipitation

Yes indeed, and we have removed this part as our dataset is not precise enough. We mainly focus on the streams' shear stresses.

- Lines 107-109. This is confusing – hot air cannot rise and is trapped against the foothills, but also cools at high altitude?

Yes, indeed. We have changed this sentence.

- Line 112. If you refer to Pisco, mark it on the map

It was referring to Piura which is outside of our study area so we have removed the sentence

- Line 143. The D96 is not the maximum particle size

This has been corrected

- Line 183. This sentence makes a big claim and needs to be supported by some key citations

Citations have been added to support the sentence.

- Line 293. Is the fracture spacing 10-20cm? Because this is the particle size range. I'm sure fracture spacing sets the sizes of large boulders, but I'm not convinced this mechanism applies to pebbles

Yes indeed. We have removed this part of the paper as it was non-conclusive.

- Line 295. The authors state that abrasion makes particles more spherical, and then say it doesn't. Please clarify which it is

We have clarified this point. As particles are transported over longer distances, abrasion tends to equalize the length of the three axes, thus making a particle more spherical. While this concept is likely to be valid for pebbles with a homogenous fabric, it likely fails to describe abrasion and break-down of material with an inherited planar geologic fabric (such a gneisses and sediments).

- Line 300. Yet the southernmost catchments in the southern grouping are very small, but show the roundest clasts. Is this not contradictory?

Yes, indeed. We have changed this part

- Fig. 5. These axes should be reversed

The figure has been removed and another figure with the same axis for every graphs

[Figure]

[Figure]

**Geology**

Multiple sampled basin
Studied basins
Sampling site
City
Western escarpment

Quaternary deposits
Tertiary rocks
Coastal batholith
Mesozoic sedimentary rocks
Metamorphic rocks

**Annual precipitations TRMM**

3000 mm/yr

0 mm/yr

**Figure 1: A:** Map of the studied basins showing the sampling sites and the western escarpment (western escarpment modified after Trauerstein et al., 2013). **B:** Geological map of the western Peruvian Andes. **C:** Map of the precipitation rates showing the spatial extend of the ITCZ, modified after Huffman et al., 2007.)

**Fig. 1.**

[Figure]

[Figure]

**Figure 2:** Geological map of the Majes basin overlain by the precipitation pattern (Precipitation data from Steffen et al., 2010., where the black dashed lines show precipitation rates (mm/yr). GS1 to GS5 represent sites where grain size data has been collected. The right corner shows the Majes river long profile.

**Fig. 2.**

[Figure]

* Data from Reber et al., in review

Figure 3: Topography of subducting Nazca plate, where slab depth data has been extracted from earthquake.usgs.gov/data/slab/. This N-S projection also illustrates: a) tectonic lineaments such as submarine ridges and MFZ: Mendaña Fracture Zone; NFZ: Nazca Fracture Zone; b) Holocene Volcanoes; c) Earthquake data, taken from earthquake.usgs.gov/earthquakes/search/; number of earthquakes M>4 within 30 km radius window; d) Coastal elevation. The data has been extracted from a 20 km-wide swath profile along the coast. The three lines represent maximum, mean and minimum elevations within the selected swath; e) Catchment averaged denudation rates have been corrected for quartz contents; f) Mean annual precipitation rates; g) Mean annual water discharge; h) Mean basin slope i)Grain size results for the intermediate (b)-axis of the pebbles in the streams from north to south at the sampling sites presented in Figure 1; j) Ratio between the intermediate axis and the long (a)-axis (modified after Reber et al., in review).

**Fig. 3.**

[Figure]

Grain size (cm)

D96 = 9.5 (± 0.3 e (-0.0108 (± 0.0003) x)
R2 = 0.92

△ D96
■ D84
○ D50

R2 = 0.97

D84 = 7.3 (± 0.2) e (-0.0085 (± 0.001) x)

R2 = 0.68
D50 = 4.8 (± 0.11) e (-0.0022 (± 0.0002) x)

Distance from the coast (km)

**Figure 4:** Grain size results along the Majes River.

**Fig. 4.**

| River name | Sample name | Altitude (m) | Latitude (DD WGS84) | Longitude (DD WGS84) | D50 (cm) | D84 (cm) | D96 (cm) | b/a | Catchment area (km2) | Mean elevation (m.a.s.l.) | Mean slope (m/m) | Slope at the sampling site (m/m) | Distance form the western escarpment (km) | Mean annual precipitation (mm/yr) (Reber et al., in review) | Mean annual water discharge (m3/s) (Reber et al., in review) | Denudation rates (mm/ka) (Reber et al., in review) | Denudation rates uncertainties (mm/ka) (Reber et al., in review) | Denudation rates corrected for O2 content in bedrock (mm/ka) (Reber et al., in review) |
|---|---|---|---|---|---|---|---|---|---|---|---|---|---|---|---|---|---|---|
| Tacna | PRC-ME1 | 231 | -18.12 | -70.33 | 2.3 | 6.2 | 10.0 | 0.70 | 899 | 2733 | 0.28 | 0.015 | 48 | 149.6 | 3.4 | 13.3 | 3.6 | 12.2 |
| Rio Sama Grande | PRC-ME3 | 455 | -17.82 | -70.51 | 2.5 | 5.5 | 10.6 | 0.67 | 2150 | 3105 | 0.3 | 0.013 | 73 | 136.4 | 4 | 28.6 | 5.3 | 27.7 |
| Ilo / Rio Osmore | PRC-ME5 | 1072 | -17.29 | -70.99 | 2.6 | 5.1 | 7.8 | 0.70 | 1783 | 3398 | 0.26 | 0.018 | 53 | 137.8 | 3.4 | 21.4 | 4.8 | 18.6 |
| Rio Tambo | PRC-ME6 | 145 | -17.03 | -71.69 | 1.5 | 3.6 | 7.5 | 0.69 | 12885 | 3568 | 0.24 | 0.051 | 141 | 216.3 | 38.1 | 89.7 | 16.7 | 72.1 |
| Tambillo / Rio Sihuas | PRC-ME802 | 117 | -16.34 | -72.13 | 2.0 | 6.0 | 10.0 | 0.69 | 1708 | 3285 | 0.15 | 0.019 | 70 | 170.2 | 30.1 | 34 | 6.4 | 27.7 |
| Camana / Rio Majes | PRC-ME7 | 69 | -16.51 | -72.64 | 5.2 | 8.7 | 11.6 | 0.67 | 17401 | 3635 | 0.23 | 0.005 | 188 | 283.9 | 68.4 | 127.5 | 23.4 | 106.8 |
| Ocona / Rio Ocona | PRC-ME9 | 14 | -16.42 | -73.12 | 4.8 | 6.8 | 10.0 | 0.71 | 16084 | 3745 | 0.26 | 0.004 | 192 | 414.7 | 91.1 | 242.1 | 45 | 184.1 |
| Nasca / Rio Grande | PRC-ME1402 | 15 | -15.85 | -74.26 | 1.3 | 3.0 | 6.0 | 0.71 | 1412 | 2716 | 0.32 | 0.014 | 48 | 283.6 | 20.4 | 46.1 | 8.6 | 29.4 |
| Chacaltana / Rio Ica | PRC-ME15 | 3 | -15.63 | -74.64 | 2.9 | 6.4 | 9.6 | 0.73 | 4677 | 2204 | 0.26 | 0.003 | 88 | 188.4 | 12.1 | 27 | 5.7 | 25.1 |
| Humay District / Rio Pisco | PRC-ME16 | 400 | -13.73 | -75.89 | 3 | 6.6 | 13 | | 3649 | 3464 | 0.34 | 0.013 | 62 | 272.6 | 13.6 | 104.1 | 20.4 | 69.1 |
| Chinca Alta / Rio San Juan | PRC-ME17 | 75 | -13.47 | -76.14 | 1.3 | 3.8 | 7.6 | 0.69 | 3090 | 3197 | 0.37 | 0.01 | 78 | 237.8 | 10.1 | 61.2 | 11.7 | 44.1 |
| Rio Canete | PRC-ME19 | 23 | -13.12 | -76.39 | 2 | 4.6 | 8.8 | 0.72 | 6029 | 3648 | 0.4 | 0.01 | 100 | 318.4 | 26.4 | 66.8 | 12.3 | 51.2 |
| Rio Omas | PRC-ME20 | 33 | -12.67 | -76.65 | 1.6 | 4.8 | 8.8 | 0.73 | 2322 | 3294 | 0.41 | 0.0076 | 78 | 257.6 | 8.2 | 27.1 | 5.4 | 17.9 |
| Rio Lurin | PRC-ME22 | 40 | -12.25 | -76.89 | 3 | 5 | 8.8 | 0.74 | 1572 | 2568 | 0.38 | 0.022 | 70 | 175.5 | 3.7 | 38.5 | 7.1 | 23.6 |
| Lima / Rio Chillon | PRC-ME39 | 402 | -11.79 | -76.99 | 5.3 | 10.5 | 15.5 | | 1755 | 2942 | 0.39 | 0.018 | 51 | 204.7 | 4.9 | 82.2 | 15.5 | 53.4 |
| Rio Chancay | PRC-ME23 | 72 | -11.61 | -77.24 | 5.5 | 8.3 | 12.5 | 0.74 | 3059 | 2697 | 0.39 | 0.01 | 66 | 211.4 | 8.9 | 97.7 | 18.4 | 52.8 |
| Rio Supe | PRC-ME25 | 74 | -11.07 | -77.59 | 2.8 | 7.7 | 13 | | 4306 | 2365 | 0.38 | 0.012 | 82 | 275.4 | 3.8 | 41.7 | 7.7 | 25.6 |
| Rio Pativilca | PAT-ME | 10 | -10.72 | -77.77 | 1.8 | 3.6 | 6 | | 4607 | 3378 | 0.44 | 0.014 | 74 | 490.6 | 30.9 | 260.1 | 48.8 | 190.9 |
| Huarmey | PRC-ME38 | 24 | -10.07 | -78.16 | 1.7 | 3.4 | 5.2 | | 2072 | 2337 | 0.37 | 0.004 | 78 | 340.1 | 9.8 | 19.7 | 4.5 | 10.1 |
| Rio Santa | PRC-ME27 | 80 | -8.97 | -78.62 | 2 | 5.4 | 9 | 0.72 | 12313 | 3262 | 0.38 | 0.005 | 65 | 571.7 | 96.1 | 71.2 | 13.4 | 70.4 |
| San Martin de Porres | PRC-ME30 | 67 | -7.32 | -79.48 | 2.9 | 6.3 | 10 | | 3882 | 2292 | 0.34 | 0.007 | 126 | 472.8 | 25.4 | 30.5 | 5.9 | 25.8 |

Table 1 : Location of the sampling sites with the altitude in meters above sea level.
The table also displays grain size results together with the rivers' and basins' properties and hydrological properties.
Morphometric dataset for the sampled drainage basins. All calculations are based on the 90 m resolution DEM (NASA)
The precipitation, water discharge data and the denudation rates are from Reber et al., in review

**Fig. 5.**

|  | Distance from the coast (km) | Altitude (m) | Latitude (°) | Longitude (°) | D50 | D84 | D96 | b/a |
|---|---|---|---|---|---|---|---|---|
| GS1 | 20 | 69 | -16.51 | -72.64 | 5.2 | 8.7 | 11.6 | 0.67 |
| GS2 | 45 | 283 | -16.37 | -72.49 | 4.8 | 10 | 15 | 0.69 |
| GS3 | 57 | 378 | -16.28 | -72.45 | 5.4 | 12.7 | 21 | 0.65 |
| GS4 | 90 | 700 | -16.00 | -72.48 | 3.3 | 12 | 22.5 | 0.67 |
| GS5 | 106 | 882 | -15.86 | -72.45 | 6.2 | 19 | 31 | 0.71 |

Table 2: Location of the sampling sites in the Majes basin and grain size results in the Majes basin.

**Fig. 6.**

| | Altitude (m) | Latitude (DD WGS84) | Longitude (DD WGS84) | D50 (cm) | D84 (cm) | D96 (cm) | b/a | Catchment area (km2) | Mean elevation (m.a.s.l.) | Mean slope (m/m) | Distance form the western escarpment (km) | Mean annual precipitation (mm/yr) | Mean annual water discharge (m3/s) | Shear stress | Denudation rates (mm/ka) | Denudation rates corrected for Qz content in bedrock (mm/ka) |
|---|---|---|---|---|---|---|---|---|---|---|---|---|---|---|---|---|
| Altitude (m) | 1.00 | | | | | | | | | | | | | | | |
| Latitude (DD WGS84) | -0.36 | 1.00 | | | | | | | | | | | | | | |
| Longitude (DD WGS84) | 0.46 | -0.97 | 1.00 | | | | | | | | | | | | | |
| D50 (cm) | 0.09 | 0.00 | -0.01 | 1.00 | | | | | | | | | | | | |
| D84 (cm) | 0.14 | 0.04 | -0.03 | **0.87** | 1.00 | | | | | | | | | | | |
| D96 (cm) | 0.18 | 0.02 | -0.02 | **0.73** | **0.93** | 1.00 | | | | | | | | | | |
| b/a | -0.30 | 0.66 | -0.71 | 0.09 | 0.00 | -0.02 | 1.00 | | | | | | | | | |
| Catchment area (km2) | -0.25 | -0.12 | 0.12 | **0.31** | 0.16 | 0.04 | -0.25 | 1.00 | | | | | | | | |
| Mean elevation (m.a.s.l.) | 0.21 | -0.38 | 0.37 | 0.03 | -0.07 | -0.03 | **-0.49** | 0.53 | 1.00 | | | | | | | |
| Mean slope (m/m) | -0.23 | 0.72 | -0.78 | -0.07 | -0.10 | -0.03 | **0.63** | -0.28 | -0.20 | 1.00 | | | | | | |
| Distance form the western escarpment (km) | -0.32 | -0.14 | 0.14 | **0.35** | 0.16 | 0.03 | **-0.33** | 0.84 | 0.37 | -0.35 | 1.00 | | | | | |
| Mean annual precipitation (mm/yr) (Reber et al., in review) | -0.43 | 0.65 | -0.61 | -0.08 | -0.16 | **-0.23** | 0.21 | 0.44 | 0.11 | 0.39 | 0.30 | 1.00 | | | | |
| Mean annual water discharge (m3/s) (Reber et al., in review) | -0.30 | 0.03 | -0.01 | 0.18 | 0.05 | -0.07 | -0.13 | 0.87 | 0.51 | -0.23 | 0.64 | 0.66 | 1.00 | | | |
| Shear stress | 0.45 | -0.11 | 0.14 | **0.23** | **0.33** | **0.39** | -0.06 | -0.21 | 0.09 | 0.06 | -0.23 | -0.43 | -0.37 | 1.00 | | |
| Denudation rates (mm/ka) (Reber et al., in review) | -0.23 | 0.04 | -0.09 | **0.34** | 0.09 | 0.00 | -0.09 | 0.56 | 0.53 | 0.12 | 0.48 | 0.50 | 0.56 | -0.07 | 1.00 | |
| Denudation rates corrected for Qz content in bedrock (mm/ka) (Reber et al., in review) | -0.22 | 0.01 | -0.04 | 0.30 | 0.06 | -0.03 | -0.17 | 0.64 | 0.57 | 0.05 | 0.54 | 0.54 | 0.65 | -0.11 | 0.99 | 1.00 |

Table 3: Results of the statistical investigations, illustrated here as correlation matrix values.
The valuess in bold show significant correlation between the grain size data and the morphometric parameters and basins characteristics

**Fig. 7.**

---

## Editor Comment (EC1) · R. G. Hilton (Editor) · 17 May 2017

Dear authors,

Thank you for submitting your work to Earth Surface Dynamics Discussions. Two reviewers assessed the manuscript and you have had an opportunity to respond to their comments. Here I summarise my recommendations and provide additional feedback on the author reply and proposed revisions. Following my own assessment, I found myself in full agreement with both reviewers. They noted that the overall idea behind the manuscript is interesting, as is the general study area. The geographical extent of the dataset is also a strength. Indeed, there are outstanding questions regarding the controls on the spatial distributions of grain size in fluvial systems, with Allen et al., (2017), Basin Research, recently highlighting a need for grain size properties to be

documented across a wider range of climatic, topographic and tectonic settings. There is recognition from the reviewers that there are interesting insights to be gained from this dataset.

However, the reviewers raised substantial issues which need to be addressed before publication. I agree with both reviewers that the manuscript needs to be much clearer in the introduction and discussion, and at present large parts are somewhat unconvincing and confusing. The comments from the reviewers offer a way to do this. Based on my own assessment and the reviewer's comments, the most pressing issues to resolve during revision are:

1. Clearer explanation and presentation on i) the processes which govern grain size distribution in river gravel bars; and ii) how these processes interact in this dataset to explain the observed patterns. The paper should to make clear what is known from the literature in the introduction, and then work these themes through the results and discussion in a systematic way. They key will be to better demonstrate the way in which they combine in this setting. Both reviewers press on this, and have specific recommendations to help here.

2. A more robust analysis of the data and consideration of its limitations (see more detailed comments by both reviewers). These include a clearer explanation of sampling sites and how comparable these are between locations, uncertainties on grain size percentiles and some explanation on the method validation.

Thank you for your reply to the reviewers, which has commented on these main points, and shows that a revised version could address the reviewers concerns and thus be suitable for publication at ESurf. When working towards a revision, please clearly reply to the main comments I mention above, and specifically explain how they have been addressed.

Other comments on the author reply which can be considered while making the revisions:

- I note you mention an addition of a paragraph on how tectonics can influence grain size and properties. It would be useful if the text here also included more in depth discussion on the role of fluvial transport (including abrasion) and the climatic factors (discharge, discharge intensity) that could influence. The reply doesn't indicate this has been included.

- There were some comments in the reply to Reviewer 1 which were unclear, mainly about the role of transience. They seemed to suggest there was additional downstream data than that presented – which if the case should be included. Nevertheless, this comment needs to be dealt with in the revision.

- Figure 3 is a good idea to provide an overview of the site characteristics and the data. However, the inclusion of data from an 'in review' publication is somewhat problematic. That secondary data seems to now be an important part of the discussion. It needs to be referred to without compromising ethics of submission, but if it is not published data the methods and approaches need to be explained herein.

- Table 3 – This is a Pearson correlation matrix, and I guess that significance is P<0.05? It is good practice to provide P values for each r value (beneath). Note that the reply mentioned 'state of the art' statistical techniques. This is a standard technique, albeit a helpful one in this context to reveal some of the patterns.

- The graphical representation of the data is important in the community which reads ESurf. I recommend that relationships between some of the key variables are still presented as scatter plots in the revised version.

- Please be more specific when explaining how specific comments have been addressed (replies to Reviewer 2). It is not that helpful to simply state comments have been addressed when the reviewer has raised a specific point which can be discussion in more detail in your reply.

- 'Contrariwise' is not a common word, and may not be clear to non-native (and indeed

native English speakers). It is used on several occasions (i.e. it is overused). Other options include 'In contrast. . ., 'On the other hand. . .'

- Please format replies so that the reviewer's text can be distinguished from the authors reply. I'm aware that the format of the text is the same on the ESurf system, so perhaps use "R1: . . ." and 'Reply: . . ."

Other comments on the original version (not covered by the reviewers comments):

The main conclusions drawn from findings are not very clear in the abstract.

240: language here needs to be more precise with regard to the statistical nature of the relationships.

241: 'frequency'

246: it wasn't clear how glacial melt plays a role here.

247: for floods, runoff magnitude (e.g. peak water discharge) and intensity are important parameters – these need to be teased apart and discussed. Are there any hydrological data or precipitation data which can be analysed from the study are to back these claims up?

252 and 326: 'worse' => 'less'

273: reference needed to support inference of fractures.

279 – clarify what mechanism - do you mean differential abrasion rates controlled by rock type – if so explain and cite relevant work.

288 and 289: 'correlation' => 'association'

Figure 3 – horizontal lines are not needed here (same on other figures). Ensure '50', '84' and '96' are subscript.

Figure 3 B and Figure 6 B– what is the shaded area? Its not necessary.

Bob Hilton, AE Durham, UK

**ESurfD**

Interactive
comment

---

## Author Response (AR1)

Dear Editor, dear referees,

Thank you very much for handling our paper. We considered the comments as very constructive and have improved the paper accordingly.

The major changes include the improvement of the introduction with a clear explanation of how tectonics and climate operate to potentially influence the grain size pattern. Based on this, we phrased a distinct hypothesis to be tested. We also we improved the methods part by adding additional information about the sampling strategy and the data collection. We have used the Pearson's correlation coefficient to obtain statistically robust correlation between our grain size data and the morphological characteristics of the basins including mean basin slope, denudation rate and basin size, and shear stresses exerted by the streams. We found distinct correlations between the grain size pattern and these variables and have framed the discussion accordingly. Therefore, we found the grouping of basins into northern and southern domains no longer as useful and have thus re-structured the paper accordingly. In summary, the major changes include:

- Presentation of a clear outline of how tectonics and climate could influence the grain size pattern, and based on this, a formulation of a distinct hypthothesis
- Presentation of more details of how we have collected and analysed the data
- Testing through state-of-the art statistical methods whether basin shape, sediment flux and streams' shear stresses have a measurable control on the grain size pattern.

We have thus re-structured the discussion part accordingy.

Please find below a point-by-point response of how we have handled the suggestions and comments. Thank you very much for your hard work.
On behalf of the co-authors

Camille Litty

**Response to Editor**

1. Clearer explanation and presentation on i) the processes which govern grain size distribution in river gravel bars; and ii) how these processes interact in this dataset to explain the observed patterns. The paper should to make clear what is known from the literature in the introduction, and then work these themes through the results and discussion in a systematic way. They key will be to better demonstrate the way in which they combine in this setting. Both reviewers press on this, and have specific recommendations to help here.

> Processes govern grain size distribution and interaction
> We have addressed this point by adding a new paragraph in the introduction, which explains how tectonics and earthquake occurrence should influence the grain size pattern, and what we expect based on this. In the same sense, we have discussed how this should imprint the grain size pattern. Based on this, we were able to phrase distinct hypotheses to be tested. In particular, earthquakes are expected to release large volumes of landslides to the trunk stream, which, in turn, is expected to yield in the supply of larger clasts. As such, we expected a positive correlation between the grain size pattern and the frequency of large earthquakes. We have tested these relationships and have not found a positive response. Introduction has been changed to address this hypothesis.
> In the same sense, the supply of larger volumes of sediment to the trunk stream is expected to shift the gravel front farther downstream, which, in turn, should be associated with a coarsening of the material along the stream. Since we have 10Be-based sediment flux data, we were able to test these relationships and have indeed found a positive one. Introduction and the entire paper have been modified to address this point. Finally, climate could influence the grain size distribution through modifications of the streams' shear stresses. Since mean water discharge data of these streams are available, we were able to test these relationships. Again, introduction and text have been adjusted and modified to address this point.

2. A more robust analysis of the data and consideration of its limitations (see more detailed comments by both reviewers). These include a clearer explanation of sampling sites and how comparable these are between locations, uncertainties on grain size percentiles and some explanation on the method validation.

Data analysis

We have used the Pearson's correlation coefficient to obtain statistically robust correlation between our grain size data and the morphological characteristics of the basins including mean basin slope, denudation rate and basin size, and shear stresses exerted by the streams. We found distinct correlations between the grain size pattern and these variables and have framed the discussion accordingly.

Data collection

We have detailed our description about where we have collected the data, and why we have collected these sites. In particular, We selected river basins situated on the western margin of the Peruvian Andes, that were generally larger than 700 km$^2$. For these systems, 10Be-based basin-averaged denudation rates and water flux data are available, thus allowing us to explore possible controls on sediment flux and shear stresses on the grain size pattern. Sampling sites were situated in the main river valleys in the western Cordillera just before it gives way to the coastal margin. We selected the downstream end of these rivers because the grain size pattern at these sites is likely to record the ensemble of the main conditions and forces controlling the supply of material to the trunk stream farther upstream. We randomly selected c. five longitudinal bars where we collected our grain size dataset. Sampling sites are all accessible along the Pan-American Highway (see Table 1 for the coordinates of the sampling sites). We have added these statements in the revised version of our paper.

- I note you mention an addition of a paragraph on how tectonics can influence grain size and properties. It would be useful if the text here also included more in depth discussion on the role of fluvial transport (including abrasion) and the climatic factors (discharge, discharge intensity) that could influence. The reply doesn't indicate this has been included.

We apologize not to have been specific enough in our previous reply letter. While we cannot fully address the potential controls of discharge intensity on the grain size pattern with the available dataset, we were able to calculate water shear stresses for mean annual runoff magnitudes of these streams, and we did see positive correlations with the grain size data. We have thus discussed these aspects in full detail and also mentioned limitations in our analyses, which are set by the available information as note above.

We could not fully discuss the controls of abrasion on our grain size data, mainly because we lack a required high-resolution dataset. Nevertheless, since the bedrock lithology is nearly constant along strike, we do not consider that abrasion has a predictive power on our grain size pattern at the scale of the western Andean margin. We do note, however, that inferred shorter transport timescales in smaller and steeper basins is likely to decrease the timescale of abrasion, which would explain the sphericity pattern we have obtained. We have mentioned these issues in our revised article, but also added cautionary notes because data on these timescales are not available.

- There were some comments in the reply to Reviewer 1 which were unclear, mainly about the role of **transience**. They seemed to suggest there was additional downstream data than that presented – which if the case should be included. Nevertheless, this comment needs to be dealt with in the revision.

We have presented and discussed all available data. We were not able to identify any transient passage of sedimentary pulses through individual basins as we have not sufficient data on downstream patterns of grain size. However, we were able to address the issue of equilibrium versus transience at the scale of the entire Western Andean margin. We have mentioned this point at the very end of the paper where we wrote: *This suggest that the ensemble of erosional and sediment transport processes have reached an equilibrium at the scale of individual clasts, but also at the reach scale of rivers where the sedimentary architecture and the clast fabric of the channel fill has dynamically adjusted to water and sediment flux and their specific time scales. Accordingly, we see the western Peruvian margin as ideal laboratory to analyze the relationships between sediment supply and water runoff on the grain size pattern of the bedload, and we propose that the bedload caliber of these streams has reached an equilibrium to environmental conditions including water discharge, sediment flux and channel geometries.*

- Figure 3 is a good idea to provide an overview of the site characteristics and the data. However, the inclusion of data from an 'in review' publication is somewhat problematic. That secondary data seems to now be an important part of the discussion. It needs to be referred to without compromising ethics of submission, but if it is not published data the methods and approaches need to be explained herein.

The publication has now been accepted in Terra Nova

- Table 3 – This is a Pearson correlation matrix, and I guess that significance is P<0.05? It is good practice to provide P values for each r value (beneath). Note that the reply mentioned 'state of the art' statistical techniques. This is a standard technique, albeit a helpful one in this context to reveal some of the patterns.

We added a Table 4 of the p-values

- The graphical representation of the data is important in the community which reads ESurf. I recommend that relationships between some of the key variables are still presented as scatter plots in the revised version.

We have added a Figure 5 presenting some of the correlations to give a graphical representation of our data. Figure 5: Correlations between the grain size data and the river parameters. A: D50 versus sediment fluxes. B: D84 versus shear stress exerted by the water .C: D96 versus shear stress exerted by the water. D: Ratio b/a versus mean catchment slope.

- Please be **more specific when explaining how specific comments have been addressed (replies to Reviewer 2).** It is not that helpful to simply state comments have been addressed when the reviewer has raised a specific point which can be discussion in more detail in your reply.

We apologize not to have properly addressed these points. We have fully re-addressed this topic. We infer that a higher seismicity, which indicates a higher degree of tectonic processes at work, results in the release of large volumes of landslides, which in turn, would shift the bedload material to coarsen. We have thus explored possible correlations between our grain size data and the frequency of high-magnitude earthquakes, but have not found any correlations.

Likewise, high surface uplift rates are expected to steepen landscapes, thereby supplying coarser grained material to the trunk streams. We have taken the occurrence of raised Quaternary terraces as proxy for most recent surface uplift and have explored whether this correlates to grain size trends. We have not found any.

We acknowledge that we have not been sufficiently careful about these issues in our previous manuscript. We took the occasion to explain more carefully our approach during this revision.

- 'Contrariwise' is not a common word, and may not be clear to non-native (and indeed native English speakers). It is used on several occasions (i.e. it is overused). Other options include 'In contrast... 'On the other hand...'

We have changed the 'contrariwise' with 'in contrast' or 'on the other hand'

240: language here needs to be more precise with regard to the statistical nature of the relationships.

This part has been removed

241: 'frequency' Done

246: it wasn't clear how glacial melt plays a role here.

This part has been removed as we lack of data

247: for floods, runoff magnitude (e.g. Peak water discharge) and intensity are important parameters – these need to be teased apart and discussed. Are there any hydrological data or precipitation data which can be analysed from the study are to back these claims up?

We have added a paragraph on the hydrological control on the grain size distribution. We have taken the precipitation rates and water discharge data form Reber et al. in press and analysed the possible correlations with our grain size data. We have also calculated the shear stress exerted by the stream water. However we note that we lack of peak discharge data and maximum precipitation intensity data.

273: reference needed to support inference of fractures.

We had no indication of the size of the fabric network. We have removed this part of the discussion as we found more compelling evidence for correlations with other variables

– Clarify what mechanism - do you mean differential abrasion rates controlled by rock type – if so explain and cite relevant work.

L 351-354: In particular, the fining rate not only depends on the abrasion (Dingle et al., 2017) and the selective entrainment processes upon transport (Ashword and Ferguson, 1989), but also on the rate at which sediment is supplied to the rivers (e.g. McLaren, 1981; McLaren and Bowles, 1985).

and 289: 'correlation' => 'association'

ok

Figure 3 – horizontal lines are not needed here (same on other figures). Ensure '50','84' and '96' are subscript.

This has been made in fig 3, 4 and 5

Figure 3 B and Figure 6 B– what is the shaded area? Its not necessary.

These figures have been modified

**Response to Referee #1**

It lacks a clear explanation of how the different factors that are meant to influence grain size operate, both in the introduction and throughout the discussion. For example, it is stated that increased uplift will be expected the increase grain size, but the causal mechanism is not described.

> We have addressed this point by adding a new paragraph in the introduction, which explains how tectonics and earthquake occurrence should influence the grain size pattern, and what we expect based on this. In the same sense, we have discussed how this should imprint the grain size pattern. Based on this, we were able to phrase distinct hypotheses to be tested.

There is also a difficulty in separating out the different mechanisms; for example, smaller basins seem to be correlated with lesser uplift, hence it is not obvious which of these two factors is more important.

> This has been confusing, indeed. We thus have completely modified the analysis plus we have framed the discussion in a different way.

Another issue is that the paper seems to alternate between assuming that downstream fining is caused by abrasion and that it is caused by selective transport, without any explicit consideration of which process is likely to be more important, or the implications of one process being dominant. (A relevant paper for the discussion of abrasion processes is Sklar et al., 2006.)

> It is true that we did not take into consideration the different processes. We have clarified this point and have been consistent in our interpretation.

Overall, I would have liked a greater sense of the underlying processes that control grain size, how they interact with each other, and the relative importance of the different factors.

> This has been done. We have rephrased the discussion section, thereby addressing the interplay between the controls of the various variables more carefully.

I also have some queries about the way in which the data were collected and analysed. The authors do not state how the locations in the different river basins were selected (other than the presence of the highway).

> This information has been added. In fact, we have sampled all streams where upstream basin sizes were larger than 700 $km^2$, and we have focussed our data collection at the downstream end where these rivers cross the tip of the mountain belt. This strategy allows us to explore how the ensemble of all processes in a basin relevant for the supply of material influences the grain size patter. This has been clarified in the revised version of the paper.

My concern is that they are attempting to compare grains sizes that are collected from different relative locations within the basin, and are therefore not comparing like with like. For example, if the basins all had the same rate of downstream fining but the samples were collected from different locations within the basins, then the analysis would show differences between the basins that are not actually there. The authors need to consider this as a possible source of variation within their results.

> This could indeed add a bias, however, we have selected streams where they cross the tip of the Andean mountain belt. Please see also comment above.

It would be useful to consider sample location as a function of total basin length, and also to normalise the distance to the knickpoint.

> We have considered this variable (distance from edge of Western Escarpment). Please see revised version. We have not performed this normalization, but used other variables instead (e.g., shear stresses, basin-averaged denudation rates), which yield measures for flow strengths and sediment flux. Because grain size and fining trends potentially depend on these variables, we used these variables for our analysis and we have indeed found positive correlations with grain size patterns.

There is also the question as to whether these basins are in a form of equilibrium or whether the grain size might actually reflect transient processes such as a coarse sediment slug progressing through the basin. I think that you need more discussion of the literature on controls on downstream grain size; at present the relevant papers are only referred to in passing at the start of the introduction.

This might work for individual basins, such as exemplified for Majes, where the grain size decreases downstream. However, this does not work if all basins along the western Peruvian margin of the Andes are considered, because the D50, as an example, increases with downstream distance from the uppermost edge of the Western Escarpment. In fact, we would have expected the opposite where grain sizes decrease with increasing transport distance. However, we found positive correlations with grain size and mean basin slope, mean basin denudation rates and shear stresses of the streams. This suggests that supply of material (higher denudation rates) and water flow strengths have a large influence on the downstream fining trends within each basin. We have thus framed the discussion in this direction.

What were the channel morphologies

The channels have a braided pattern, and the morphology of the longitudinal stream profiles is characterized by two segments  separated by a distinct knickzone. Please see revised version.

how large were the individual images ?

This has been clarified: Individual images are about 1 m$^2$.

how were grains selected within the images ?

Every pebble, which was entirely visible on the digital images, has been measured.

how representative are the selected bars ?

For these basins, sampling sites were situated in the trunk streams of these valleys where the streams cross the tip of the mountain belt, which is located near the Pacific Coast in most cases. We selected the downstream end of these streams because the grain size pattern at these sites is likely to record the ensemble of the main conditions and forces controlling the supply of material to the trunk stream in the upstream basin and thus the grain size caliber of these streams where they leave the Andes. In these streams, we randomly selected c. 5 longitudinal bars where we collected our grain size dataset. As such, we consider the selected bars as representative for the ensemble of supply and transport processes in the streams' basins.

how were grain outlines identified (automated or manual analysis) ?

From those photos, the intermediate b-axes and the long a-axes of around 500 pebbles were manually measured. We have added this information in the revised version.

was any attempt made to verify the grain size data produced,

No attempt has been made to verify the grain size data produced for this paper. Nonetheless, all the pebbles have been measured by the same operator. This yields the same bias for every sampling site, if there is any.

why were 500 extra grains used for grain shape

We have clarified this in the method part and what are the error bars on D50/D84/D95 (and hence are the identified differences significant)?

> Uncertainties on the grain size percentiles are also about 3 mm. This value corresponds to the precision limits of the measurements with the software ImageJ and of the digital pictures' resolution. For the significance of the difference, correlations or trends have all been estimated through the Pearson correlation coefficient (p-value) and not anymore on a visual estimation as we have don before.

The lack of a clear hypothesis early on means that some of the analysis comes across as a bit of a fishing expedition, with lots of correlations on different data groupings being undertaken, and only the significant ones being presented. I think that you need to be more thorough about this analysis, for example through multiple or stepwise regression.

> This has been done, and a hypothesis has been phrased. Please see also comment above.

Comments by line: 10: Overall the abstract could be more specific and provide some more evidence for the various claims.

> We have addressed this point

53: To what extent are these different factors interrelated?

> We have addressed this point by adding a new paragraph in the introduction

55: Make it clearer how this information about the general setting is related to the overall aim of understanding grain size.

> Done

78: Be more explicit about why uplift produces larger clasts.

> This has been specified

79: You describe both N-S and E-W variations; which are most important for your study?

> N-S are more relevant; we have rephrased the introduction and clarified this point.

97: I'm surprised that erosion is nearly zero (line 89) given this high precipitation.

> Abbühl et al. (2011, ESPL) have shown that the low denudation rates are due to the flat landscapes on the Altiplano.

122: Be more specific about uplift rates.

> This has been changed accordingly

125: Is five sites enough to identify trends?

> We have changed our data interpretation as there was no real reason to separate the basins into 2 groups (i.e. northern and southern domains). We have worked with all our dataset.

196: Calculate sorting parameters to quantify these trends.

> No trend actually exists as there is no real change in the grain size from north to south, so we also did not introduce the sorting parameter.

176: Suggests that you are downstream of the gravel-sand transition? Does the transition occur in other basins?

This transcription seems to not occur in the other basins. We have addressed this point in the discussion part.

195: It would be useful to calculate stream power, as this would enable you to look at the combined impact of slope, width and discharge.

Yes, indeed, but we have calculated shear stresses instead. We have done so and we do see correlation in between the grain size and the shear stress.

201: Is the relationship significant?

Indeed, we are not considering anymore the grouping (northern and southern domains) of basins.

209: Overall there are many competing ideas in the discussion, and it's not clear which are most important.

This has been confusing, indeed. We thus have completely modified this part of the analysis plus have framed the discussion in a different way.

216: This is the first mention of sediment sources; this needs to go earlier in the paper.

We now mention it earlier in the text. 'The upstream edges of this knickzone called the Western Escarpment also delineate the upper boundaries of the major sediment sources'

225: Note that rivers can also adjust to changes in uplift by changing other factors such as width, morphology and the amount of sediment cover.

Yes we have changed the part on the tectonic control on grain size

244: What is the mechanism that relates different flood characteristics to different grain sizes?

We have rephrased the entire discussion and have likewise changed this section.

257: What is your evidence?

Because we have only found a correlation between the D50 and the basins scale properties (basin area, denudation rates, mean slope, we infer that the mean grain size reflects the ensemble of a complex pattern of erosional processes operating in the Peruvian basins

273: How does the size of this fracture network compare to the grain sizes?

We had no indication of the size of the fabric network. We have removed this part of the discussion as we found more compelling evidence for correlations with other variables.

287: This argument would be stronger if you presented the lithological characteristics of your grains, which you could identify from the photos.   Or state that they are all identical within each basin. 300: Is this consistent with the geological variations?

A test of the inferred positive correlation between mean basin slope, bedrock lithology and particularly the occurrence of plutonic rocks, and the pebbles' sphericity would require a higher resolution topographic and geologic data, which are currently not available, we thus decided to remove this part, which also does not fit anymore in the discussion, as the grouping of basins into southern and northern domains is not considered anymore.

288: Note that you only have information on 2D grain shape not 3D.

Yes indeed, these are the a- and b-axis. So we are indeed missing the information about the third dimension to talk about the shape of the clasts. In this sense, the reviewer is correct. Nevertheless, we are still convinced that the 2D info contains valuable information about the shape of the clasts in the sense that preferential abrasion due to an inherited fabric (fractions, bedding, schistosity) returns elliptical rather than spherical clasts. We have thus kept this part of our analysis.

296: Which idea do you think is more correct?

This point has been addressed in the revised version of the text.

321: I'm still not entirely clear what you mean by a 'geomorphic' control.

It was indeed unclear, we have rephrased that. But what we wanted to say is that the geomorphic parameters (basin slopes, size, denudation) were controlling the grain size distribution

323: But much of the earlier discussion has referred to abrasion.

We have indeed been contradictory. However, we have substantially changed the paper and thus also the conclusions.

Table 1: Add an indication of where the site is relative to the knickpoint and within the basin.

This has been done in the method part

It would help to also present distances normalise by total basin length.

We did not normalize by the basin length because this is one of the parameters that we wanted to test as control on the grain size

Table 3: Give sorting values.

We have deleted table 3 as we do not group the basins into northern and southern basins

Figure 1: Add basin outlines to maps B and C.

This has been made

Figure 2: Add the channel.

This has been made

**Response to Referee #2**

The authors rule out a tectonic control by simply stating that greater surface uplift rates should result in larger clast sizes. Why?

We have fully re-addressed this topic. We infer that a higher seismicity, which indicates a higher degree of tectonic processes at work, results in the release of large volumes of landslides, which in turn, would shift the bedload material to coarsen. We have thus explored possible correlations between our grain size data and the frequency of high-magnitude earthquakes, but have not found any correlations.

Likewise, high surface uplift rates are expected to steepen landscapes, thereby supplying coarser grained material to the trunk streams. We have taken the occurrence of raised

Quaternary terraces as proxy for most recent surface uplift and have explored whether this correlates to grain size trends. We have not found any.

We acknowledge that we have not been sufficiently careful about these issues in our previous manuscript. We took the occasion to explain more carefully our approach during this revision. In particular, we wrote: *Among the various conditions, hillslope erosion and the supply of material to the strunk stream has been shown to mainly depend on: (i) tectonic uplift resulting in steepening of the entire landscape (Dadson et al., 2003; Safran et al., 2005; Wittmann et al., 2007; Ouimet et al., 2009), (ii) earthquakes and seismicity causing the release of large volumes of landslides (Dadson et al., 2003; McPhillips et al., 2014), (iii) precipitation rates and patterns, controlling the streams' runoff and shear stresses (Litty et al., 2016), and (iv) bedrock lithology where low erodibilty lithologies are sources of larger volumes of material (Korup and Schlunegger, 2009),. Because most of the bedload material of rivers has been derived from hillslopes bordering these rivers, as mapping and grain size analyses of modern rivers in the Swiss Alps have shown (Bekaddour et al., 2014; Litty and Schlunegger, 2017), it is very possible that the grain size distribution of modern rivers either reflect the seismic processes at work, or rather reveal the response to the climate conditions such as rainfall rates and the shear stresses of rivers.*

The mechanism underpinning this assumption (e.g., enhanced landsliding as a result of incision, etc) is very important if you want to look for tectonic signals in sedimentological data.

Yes indeed. However, we mainly focussed on the frequency of earthquakes, which should influence the occurrence of landslides and thus the grain size pattern of streams. We have selected this approach because earthquake data were available.

The Methods section requires more information about where the grain size were collected.

Improved and largely expanded.

'Along a highway' isn't very helpful – were the measurements made at equivalent locations in the longitudinal profiles of the catchments?

Yes indeed. We have outlined more carefully why we have selected our basins, and where we have done the measurements.

If you want to compare measurements from one catchment to another, it's important to demonstrate that the data come from comparable sampling sites.

Yes, indeed. We acknowledge that we have not properly explaine our sampling strategy and have now specified this point.

It would also be helpful to know where the discharge data were collected in the catchments. I appreciate that the coordinates are listed in Table 1, but some description is needed about whether the discharge data represent equivalent points in the catchments; i.e., if one catchment is sampled at the mouth and another at the headwaters, how can a meaningful comparison be made?

We have taken the discharge data from Reber et al., in press in Terra Nova. These authors provide the full information about the data source.

I have some major criticisms of the results. Uncertainties are needed on the grain size percentiles, because the scatter in Fig. 3a is larger than the trends the authors interpret.

The grain size data have too large a scatter, so interpretations of trends are indeed not possible. We have changed the manuscript accordingly. We have worked on statistical correlations using the Pearson's coefficient and no correlation has been found between the D50 and the latitude. Uncertainties on the grain size percentiles are also about 3 mm. This value corresponds to the precision limits of the measurements with the software ImageJ and of the digital pictures resolution.

The way the authors describe the grain size data from line 162 onwards implies a systematic variation from north to south, which is not really true.

Indeed, we have changed the analyses, and there is indeed not such a trend.

It should be clarified that the rates of grain size change from north to south refer to an average regression fitted to the data.

We have changed the analyses, and there is not such a trend.

The whole paragraph from line 171 is not really a description of results, and could be moved to the Discussion.

We have changed the analyses, so this paragraph has been removed.

However the final point (line 176) is very important and needs some explanation.

We have added an all paragraph on the gravel front in the discussion

Why are there catchments in the middle of the study area that apparently have much bigger grain size differences (only sand and no gravel) than the catchments examined in the paper?

We have discussed this point.

The authors are apparently aware of much larger grain size variability in the area but have ignored those catchments, and it is not clear to me why.

We have discussed this point.

There are some issues with Fig. 3. The data in panel A are compressed to the bottom of the graph and half the plot isn't used – please expand the data so the reader can better see the trends (the annotations can go above the graph).

In panel B, I am concerned that some of the data points are missing between 5-15 degrees latitude. Why are there only 6 points (compared to 11 in A)?

Figure 3 has been improved. The ratio b/a has not been measured at each sampling site so there are less data points in the ratio plot than in the percentiles one.

Also, which percentile has been used to calculate the a/b ratio?

There is no percentile used. We measured the length of the a-axis and the b-axis were per pebble. This gives us one value for the ratio. We repeated this for 500 pebbles, yielding a mean value per sampling site.

Next, it appears the coarsest grain sizes from the northern group of catchments are being exported from the shorter catchments that only drain west of the western escarpment. Those with larger upstream reaches crossing the western escarpment have equivalent grain sizes to the southern catchments. This difference is quite apparent by comparing Figs 1 and 3, and may invalidate the north/south grouping of catchments.

Yes indeed; we also realized that and have rewritten the discussion part of the paper.

The final part of the results contrasts Figs 5 and 6.

Yes indeed. We have completely changed this part of the analysis

The authors suggest that there are no correlations between grain size and the chosen parameters in Fig. 5, but that there are correlations when the catchments are grouped (Fig. 6).

We have changed this part of the analysis

This isn't really a comparison, because the two figures are showing different things. I cannot tell how Fig. 5e and 5f would compare to Fig. 6 if the same normalisation was performed on discharge.

Indeed, please note that we have changed this part of the analysis

Why was discharge normalised in Fig. 6a but not elsewhere in the paper?

And why have the authors chosen those particular grain size percentiles and variables in Fig. 6?

This has been changed and corrected. Indeed, this did not make sense.

It seems they have simply plotted everything against everything else and shown two unrelated correlations that are not particularly convincing and do not test a particular hypothesis.

We have framed our paper around a hypothesis. So this aspect has been changed. We have made a new figure showing the data from south to north and a correlation matrix using the Pearson's correlation coefficient to give statistically robust analyses.

I am confused about why the southern catchments should be characterised by comparing runoff normalised by area with D50, while the northern catchments should be characterised by their gradients as a function of D96.

We have changed this part of the analysis

The Discussion attempts to address some important questions about grain size patterns observed in river networks and how they might record various forcings. Unfortunately, it is inconclusive and unclear. The authors claim around line 219 that fluvial transport dominates the Majes basin – if so, why does the D50 not fine over a 100 km distance?

We have addressed this point.

In section 4.2, do the arguments here require that smaller rivers in smaller basins are moving coarser material? This needs to be clarified.

We have changed this part of the analysis

For section 4.3, what is the actual difference in climate between the northern and southern domains?

We have changed this part of the analysis; we no longer perform this grouping.

In Fig. 1c, apart from the wetter patch near Huaraz (which actually overlies a catchment exporting finer grain sizes!), the two areas look similar. I recommend the authors plot the runoff data and/or precipitation against latitude (following Fig. 3) if they want to argue there is a relationship here.

We have changed this part of the analysis

They need to show that the two domains are actually different and that climate correlates with grain size if they want to make that argument.

We changed this discussion accordingly

In section 4.4, the authors could clarify whether the smaller catchments in the northern group were glaciated as well, or only the larger ones? Because the coarsest data seems to only come from the smaller catchments, and this is an important difference that needs to be addressed.

We have clarify this point

These smaller catchments also drain proportionately more of the Coastal Batholith, which might indicate an erodibility control on grain size.

We have changed this part of the analysis and the discussion accordingly.

The arguments in this section are vague and undeveloped and jump from glaciers to lithology without offering any precise interpretations.

Yes, indeed. We have removed this part as it was non-conclusive.

- "Contrariwise" is an unusual word, and I recommend using something like "on the contrary" instead

    We have learned this word from an English native speaker, so we have kept it.

- Refer to "El Niño", not "the El Niño" or "the El Niño effect" (it is not an effect). Also, on line 114 you equate El Niño with ENSO – they are not exactly the same thing. El Niño is one phase of ENSO and brings particular weather patterns, but ENSO refers to the overall oscillation between El Niño, neutral, and La Niña states in the tropical Pacific

    Yes, indeed. We have removed this part.

- "Strong precipitation rate" implies a high intensity of precipitation, which is quite different to a greater overall amount of precipitation

    Yes indeed, and we have removed this part as our dataset is not precise enough. We mainly focus on the streams' shear stresses.

- Lines 107-109.  This is confusing – hot air cannot rise and is trapped against the foothills, but also cools at high altitude?

    Yes, indeed. We have changed this sentence.

- Line 112. If you refer to Pisco, mark it on the map

    It was referring to Piura which is outside of our study area so we have removed the sentence

- Line 143. The D96 is not the maximum particle size

    This has been corrected

- Line 183. This sentence makes a big claim and needs to be supported by some key citations

    Citations have been added to support the sentence.

- Line 293.  Is the fracture spacing 10-20cm?  Because this is the particle size range. I'm sure fracture spacing sets the sizes of large boulders, but I'm not convinced this mechanism applies to pebbles

    Yes indeed. We have removed this part of the paper as it was non-conclusive.

- Line 295.  The authors state that abrasion makes particles more spherical, and then say it doesn't. Please clarify which it is

    We have clarified this point. As particles are transported over longer distances, abrasion tends to equalize the length of the three axes, thus making a particle more spherical. While this concept is likely to be valid for pebbles with a homogenous fabric, it likely fails to describe abrasion and break-down of material with an inherited planar geologic fabric (such a gneisses and sediments).

- Line 300. Yet the southernmost catchments in the southern grouping are very small, but show the roundest clasts. Is this not contradictory?

    Yes, indeed. We have changed this part

- Fig. 5. These axes should be reversed

    The figure has been removed and another figure with the same axis for every graphs

[revised manuscript text omitted]

---

## Author Response (AR2)

Dear Editor, dear Referee,

Thank you very much for handling our paper once again. We have improved the paper according to the very constructive and helpful comments by yourself and the reviewer.

Following your suggestions, we have seen that values of the D50 are indeed nearly constant along the entire western Peruvian margin and range between 2 and 3 cm. The largest D50 with values up to 6 cm have been measured in streams that are either sourced in the Cordillera Negra where mean basin slope angles are larger than 20°, or in the Rio Ocoña and Rio Camaña rivers located at 16°-17°S, which have the largest mean annual discharge as they capture their waters from a broad area on the Altiplano. We thus suggest that the generally uniform grain size pattern has been perturbed where either mean basin slopes, or water fluxes exceed threshold conditions.

The major changes include the improvement of the discussion part with a focus on the comparison between the particularly larger D50 and the basins where hillslope gradients are steeper than 0.4 on the average (i.e., 20-22°), or where mean annual stream flows exceed the average values of the western Peruvian streams (10-40 m3/s) by a factor of 2. In addition, we have updated the Figures with more information about earthquake occurrence, and we have corrected the text following the Referee's recommendations. Finally, we have tuned down most of the inferences and interpretations, which were based on weak correlations only. We have thus re-structured the discussion part accordingly.

Please find below a point-by-point response of how we have handled the suggestions and comments.

Thank you very much for your hard work.

On behalf of the co-authors
Camille Litty

**[1] Comments by section:**

**Editor's comments**

Abstract:

We have tuned down the interferences and removed the statements, which lack a significant correlation. We have also removed the linkage to the earthquake occurrence, as this appears to be weakly introduced, as noted by the Editor.

Introduction:

We have improved the presentation of the past efforts in exploring the controls on the grain size pattern in streams. We made an effort to frame this section in a more global perspective, as required.

Results:

We have mentioned that the morphometric variables and related to this, information about sediment fluxes and water discharge are inter-related, as required.

We have removed the section, which discusses streams that have no gravel bars, since we have no information about where the gravel front is situated. We thus follow the recommendation by the editor not to spend too much space on an issue which lacks any data.

We indeed consider earthquake intensity as an important variable, and magnitude thresholds > 5.5 might need to be exceeded for the release of large volumes of landslides, as noted by the Editor. However, Figure 1 by Keefer (1984) suggests that earthquakes with magnitudes 4.5 are are theoretically able to release landslides over an area >10 km$^2$, which is substantial. Therefore, we have decided to keep our threshold where we considered earthquakes with a magnitude M>4.5.

We have completely rewritten the remaining part of the discussion following the recommendations by the Referee, thereby focusing on the pattern of the D$_{50}$ as recommended.

**Referee's comments**

Tectonics and geological setting:

Section 1.1 provides some good information about the geological context, but more is needed. Firstly, I am confused about the boundaries of the tectonic domains. The authors describe a change in the degree of interseismic coupling between north and south, specifically with high coupling and high seismicity south of 13-16 °S, and low coupling and low seismicity to the north. The authors cite a paper by Nocquet et al. (2014), but this paper does not contain data as far south as 13-16 °S, and instead appears to place the gap in seismicity at 3-10 °S. Almost all of the catchments in this study area are south of 10 °S, so it needs to be clarified where this boundary is. I'm struggling to reconcile the data in Nocquet et al. (2014) with the statements in this section. This needs to be cleared up.

Indeed, this paper by Nocquet does not contain data as far south as 13-16 °S. We have corrected this point and modified the tectonic section, plus we have expanded the figures showing the depth of the Nazca plate beneath the South American plate plus the frequency of earthquake occurrence over a broader scale.

Second, Fig. 3 shows a long-wavelength feature in the slope data (panel G), from 7 to 14 °S; what is this? It has quite a significant amplitude, with slopes almost doubling in the centre, but it isn't discussed. This could be significant – see my later suggestions.

We have discussed this large wavelength pattern of mean basin slopes. Please see the revised version where we wrote: *The pattern of mean slopes per drainage basin reveals a distinct N-S trend (Table 1). The corresponding values increase from 20° to 25° going from 6°S to 10°S latitude (where they reach maximum values between 0.4 to 0.45 m/m) after which they decrease by nearly 50% to values ranging between 10° and 15°. These relationships have not been explored yet, but most likely reflect the extent to which streams have crossed the western escarpment and sourced their waters in the relatively flat plateau of the Puna region. Indeed, most of the western Peruvian streams have their water sources on this flat area and then cross the western escarpment, which yields relatively low mean basin slopes particularly for basins south of 12°S. Contrariwise, the basins around 11°-12°S latitudes (which are characterized by the steep slopes) have their sources in the relatively steep Cordillera Negra (Figure 1A), which is a relatively dry mountain range situated on the steep escarpment. Along these latitudes, the high Andes are constituted by the high and heavily glaciated Cordillera Blanca situated farther to the east (Figure 1A). This mountain range is drained by the Rio Santa, which flows parallel to the Andes strike within the valley of the Rio Santa, and then crosses the Cordillera Negra at a right angle (Figure 1A).*

In this context: We agree that exceptionally larger D50 values of 4-6 cm were measured for basins situated between 11-12°S and 16-17°S where hillslope gradients are steeper than 0.4 on the average (i.e., 20-22°), or where mean annual stream flows exceed the average values of the western Peruvian streams (10-40 m3/s) by a factor of 2. We have changed the discussion part quite significantly based on the Referee's comment the reviewer have made.

Finally, what timescale do the historical earthquake records average over? Is this sufficient to compare with grain size data?

    The historical earthquakes that have been listed on the Figure 3 records earthquakes for at least the last century. In Peru, the older earthquakes that have been recorded by the USGS survey catalogue dates to the 28[th] of Sept. 1906, located in northern Peru. When it is know that sediments are evacuated on the basis of several years (one year in the Alps for example), more than a century should be a sufficiently long timescale.

Section 4.1.2. "Absence of gravels in rivers between 15.6 °S and 13.7 °S:
I'm not convinced by the arguments in this section. The authors propose that this sub-group of catchments lack gravel bars potentially because they have been lengthened by tectonic deformation, which resulted in a pulse of uplift and enhanced erosion, and the gravel-sand transition simultaneously migrated upstream into the catchments. Do the authors expect the gravel-sand transition to migrate upstream in response to enhanced erosion? This seems counterintuitive to me, and also contradicts the interpretations they make about grain size correlating (weakly) with sediment flux (e.g., Fig. 5a). Also, Fig. 4 shows the fining rates are fairly low in this landscape, with D50 only decreasing by a few mm over >100 km. How far upstream do the authors propose the gravel-sand transition has migrated in order to supply only sand at their sampling location? How much have these catchments supposedly been lengthened? As it stands, this section raises more questions than it answers.
I recommend the authors take a look at Lamb, M.P. and Venditti, J.V., 2016, The grain size gap and abrupt gravel-sand transitions in rivers due to suspension fallout, Geophysical Research Letters, 43, doi: 10.1002/2016GL068713. This paper discusses gravel-sand transitions and suggests they arise because of changes in bed shear velocity (the wash load hypothesis). Are there any reasons why these catchments might have different bed shear velocities? Perhaps the framework in this paper can help the authors develop a more robust argument, if they do want to invoke gravel-sand transitions here. At the moment this section seems contradictory and unintuitive.
A more minor point: this argument is repeated all over again in section 4.1.3 from lines 318-323, and this repetition is not needed.

    We removed this aspect of our analysis as this section was raising more questions than it answered and since the discussion about the lack of gravels was not the focus of the paper.

Section 4.1.4. "Supply control on the grain size pattern".
This section overstates some of the results. Lines 329-331: "the positive correlation between the size of the D50 and the morphometry of these basins" – there is no correlation between D50 and slope, so do the authors mean with catchment area? This is still a weak correlation, so I think this statement is over-selling the relationship. Especially given the authors then claim "environmental factors exert a major control on the pattern of the D50 encountered for the rivers in western Peru". These are weak correlations, not evidence for "major control".

    Yes, indeed, correlations are weak and our previous statements were a tentative effort to explain our dataset, but we have probably over-interpreted these weak correlations. Please see the revised manuscript where we wrote: *We consider the correlations between the grain size data (e.g., $D_{50}$) and the basins scale properties (basin area, mean basin denudation rates, water shear stresses, sediment fluxes) as not strong and convincing enough for the identification of potential controls of these variables on the grain size caliber.*

*Instead, we follow the recommendation by the Referee and focused on the pattern of the $D_{50}$ only.*

Also lines 332-335: "it is very likely that the bulk supply of hillslope-derived sediment to the trunk stream increases with larger basin size, mean basin slope and basin-averaged denudation rate" – there seems to be no correlation between D50 and mean slope, so how can this explain the grain size patterns? The relationships that can be drawn from the data are being overstated.

*Yes, these correlations are weak and relationships were overstated. Instead, we focus on the sections where the D50 were larger than on average. We have thus completely changed the discussion as recommended by the Referee.*

Next, the sentence from lines 341-344 needs to be clearer. Are the authors suggesting that the catchments with coarser grain sizes experience El Niños and extreme rainfall events with a greater sensitivity than the catchments with finer grain sizes? I find this to be very unlikely, not to mention the lack of correlation between sediment flux, water discharge and any of the grain size percentiles.

*With a lack of correlation between sediment flux, water discharge and grain size distribution, this part was overselling our data. We have refocused our analysis on the pattern of the $D_{50}$. Please see also our responses above.*

Finally, the paragraph from lines 345-364 isn't very clear either. My understanding is that the authors propose the coarser-grained catchments are experiencing a downstream shift in the position of gravel fronts, due to greater sediment supply from hillslope sources. However Fig. 5a shows that catchments can have a large D50 with either a very low sediment flux or a very high sediment flux (the full range), and Table 3 shows that no grain size percentiles correlate with catchment slopes. The authors imply that denudation rates act as a proxy for the amount of hillslope-derived sediment, but this isn't necessarily the case.
As this section was raising more questions than it answered and that it was not the focus of the paper, we removed this part.

*We have removed this entire section since it has opened more questions than it has offered answers. Please see also our response above.*

Furthermore, it might be that slopes act to perturb sediment flux (and grain size) above thresholds, e.g., some threshold angle for landslides and debris flows. In this case a simple correlation (like Table 3) might not reveal whether there is a threshold-controlled grain size response to hillslope processes, because below the threshold you wouldn't even expect a correlation.

*Yes this is what we state in this new version. This is summarized in the abstract: Exceptionally large D50 values of 4-6 cm were measured for basins situated between 11-12°S and 16-17°S where hillslope gradients are steeper than 0.4 on the average (i.e., 20-22°), or where mean annual stream flows exceed the average values of the western Peruvian streams (10-40 m3/s) by a factor of 2. We suggest that the generally uniform grain size pattern has been perturbed where either mean basin slopes, or water fluxes exceed threshold conditions.*

Section 4.1.5. "Hydrological control on the grain size distribution"
Line 373-374. "grain sizes correlate with the shear stress values". The relationships in Fig. 5b-c look quite tenuous to me.

*Yes indeed, correlations are week. The mechanisms by which grain size can be mediated through a threshold effect upon transport are less well understood, but it has been known at least since the engineering work by Shields (1936), and particularly by Peter Meyer Müller*

*(1948) that threshold conditions have to be exceeded upon the transport of grains in fluvial streams. As a consequence, at transport-limited conditions, sediment flux, and most likely also the caliber of the transported material, depends on the frequency and the magnitudes at which these thresholds are exceeded rather than on a mean value of water discharge (Dadson et al., 2003). This might be the reason why values of water shear stresses, that are calculated based on the annual mean of water flux, are not sufficiently strongly correlated with the $D_{50}$ values to invoke a strong controls thereof.*

If you took away the one point with very large grain size the correlations might even become negative?

Yes, we note that these correlations are weak and some might even break apart if the largest values (for e.g., shear stresses) are removed.

I think the text is overselling the data here, and this section is unsatisfying because [1] the plots in Fig. 5 are not conclusive, and [2] because the previous section suggested that grain size is limited by hillslope sediment supply. It's difficult to explain the data as both supply-limited and transport-limited at the same time. Also, the average particle size apparently isn't correlated with either water discharge or shear stress, so I'm really not convinced at all that it's possible to infer "a hydraulic control on grain size distribution of the Peruvian rivers" as the authors claim.

We interpret the data to point towards transport limited conditions, where sediment flux, and most likely also the caliber of the transport material, depends on the frequency and the magnitudes at which thresholds upon transport are exceeded rather than on a mean value of water discharge (Dadson et al., 2003). This might be the reason why values of water shear stresses, that are calculated based on the annual mean of water flux, are not sufficiently strongly correlated with the D50 values to invoke a strong controls thereof. The hydraulic forcing has a control only when a threshold is exceeded. We have outlined these relationships and a possible interpretation.

Section 4.2. "Transport distance and slope angle controls on sphericity"

Lines 394-398. Recycling particles from a terrace surely won't make them more spherical, because while the particles are being stored they aren't being abraded. Lines 405-406. The authors propose that in catchments with steeper slopes, denudation rates will be faster and transport distances will be shorter. Table 3 shows no correlation between slope and denudation rate (or sediment flux). This may be because slopes influence denudation rates non-linearly (see my earlier comment about the potential role of thresholds in this landscape), but either way this section seems to contradict the author's use of correlations and their data. Furthermore, it is unclear to me how steeper slopes will reduce the transport distance of material. Residence time yes, but not distance.

See 'Slope angle controls on sphericity': The relative poor positive correlation between the sphericity of the pebbles and distance from the escarpment edge prevents us from inferring a distinct control of this variable. We have thus corrected this point. Contrariwise, the positive Pearson correlation between the sphericity of the pebbles and the mean basin slope is quite high, thus pointing towards a significant control. This suggests that basins with steeper slopes, as is the case for the Cordillera Negra, produce rounder pebbles. We tentatively infer that time scales of transport and evacuation of material is likely to be shorter in steeper basins compared to shallower ones. This might influence the shape of pebbles as they tend to flatten in response to effects of abrasion and 3D heterogeneities of bedrock that becomes more obvious with time and transport distance (Sneed and Folk, 1958). We thus see the positive correlation between mean hillslope angle and the sphericity of pebbles as a very likely consequence of shorter transport times in steeper basins, but we note that this hypothesis needs to be confirmed by
detailed real-time surveys of material transport from sources down to the end of these rivers.

[2] Minor comments:

Lines 75-81. This sentence is very long and needs to be broken up. It also sounds like the authors
expect grain size to be limited by all different factors at once. If "grain size… reflects the ensemble of
mechanisms at work", then all of those mechanisms are limiting grain size together, e.g. the grain size
characteristics are both supply-limited and transport-limited at the same time. The authors should
think about whether this is really what they mean.

Line 92. Trujillo isn't marked on the map. As was suggested in the first round of review, it would be
helpful to refer to towns that are marked on the map.

Trujillo has been added on the Figure 1.

Line 93. "…up to 100 km broad coastal forearc plain". Is this 100 km wide? Long?

It is a 100 km wide broad coastal forearc plain, this has been added.

Line 120. "Andean" should be changed to "Andes".

Done.

Line 140. As was pointed out in the first round of review, "strong precipitation rates" implies a high
intensity of rainfall, which isn't the same as a greater overall amount. Consider "high precipitation
rates" instead.

We have used high precipitation rates.

Line 144. "to reach" should be "from reaching".

Done

Line 148-149. Again, like in the first round of review, please be careful about equating El Niño with
ENSO. El Niño is one state of the ENSO, but they're not the same thing.

We have carefully changed this.

Line 161. "We will use" – I recommend sticking to one tense here, present is probably best. See also
line 163, "sampling sites were situated" – use the present tense, because the sites are still there.

The sentences have been checked to stick to only one tense.

Lines 164-166. Again, see my point for lines 75-81, which also applies here. Do the authors expect
grain size to record all environmental "conditions and forces" at the same time? Put another way, is
grain size simultaneously supply-limited and transport-limited?

We have changed that in the introduction

Line 187. Percentiles should be plural.

Done

Line 204. The authors refer to a paper by Reber et al. (in press) – I still think it would be good to
briefly say how much time the discharge records cover (1 year, 100 years? Which years, i.e. are they
biased by El Niño?), and in general terms where the stations are (e.g., near the catchment mouths or
higher up in the catchments, are they are in similar places in each catchment?). It doesn't need to be a
detailed account of every station, but very briefly the reader needs to know, in this paper, whether the
data record a meaningful period of time and can be fairly compared between catchments. This only
needs 1 or 2 sentences.

An explanation has been added to the text in the methods part: *The mean annual water fluxes
were obtained by combining hydrological data reported by the Sistema Nacional de
Informacion de Recursos Hidricos (2 to c. 20 years of record) and the TRMM-V6.3B43.2
precipitation database (Huffman et al., 2007).*

Lines 208-211. This sentence is really unclear. Sediment flux can indeed affect grain size fining rates, but why does this principle mean denudation rates are variable in this study area?

We have considered the 10Be-based basin mean denudation rates (Reber et al., 2017; Table 1) as variable because in the supply-limited case, higher denudation rates could be associated with the supply of more coarse-grained material to the trunk stream, which in turn could result in larger clasts in these streams.

Line 232. "at 106 km river upstream" – this wording can be improved.

The wording has been improved: *The $D_{50}$ percentile of the b-axis decreases from 6.2 cm to a value of 5.2 cm c. 80 km farther downstream.*

Line 233. "for the Pacific coast" should be "from the Pacific coast".

Done.

Line 312. "Infer" means to deduce something, but here the authors are speculating. I would change to "expect" or something similar.

Done.

Line 313. "Increase *in* earthquake frequency".

Done.

Line 399. The 2016-2017 winter was not really an El Niño. There were restricted temperature anomalies that are sometimes called a "coastal El Niño", but this is not the same as an actual El Niño, e.g., the Niño 3.4 box showed neutral anomalies. The warm water only pooled around southern Ecuador and northern Peru.

We have changed our discussion part.

Lines 421-427. This really isn't "unravelled" in this paper. Wouldn't flattening the fabric of a clast reduce the c-axis, which isn't measured here? Either way, there's no explanation for the lack of correlation between slope and sediment flux or denudation rate, which seems counterintuitive given that sediment flux should relate to the residence time of clasts. This is more a hypothesis than a conclusion.

Indeed. We have changed the interpretation: *The relative poor positive correlation between the sphericity of the pebbles and distance from the escarpment edge prevents us from inferring a distinct control of this variable. Contrariwise, the positive Pearson correlation between the sphericity of the pebbles and the mean basin slope is quite high, thus pointing towards a significant control. This suggests that basins with steeper slopes, as is the case for the Cordillera Negra, produce rounder pebbles. We tentatively infer that time scales of transport and evacuation of material is likely to be shorter in steeper basins compared to shallower ones. This might influence the shape of pebbles as they tend to flatten as effects of abrasion and 3D heterogeneities of bedrock that    becomes more obvious with time and transport distance (Sneed and Folk, 1958). We thus see the positive correlation between mean hillslope angle and the sphericity of pebbles as a very likely consequence of shorter transport times in steeper basins, but we note that this hypothesis needs to be confirmed by detailed real-time surveys of material transport from sources down to the end of these rivers.* Please see also our response above.

Line 427-428. "the ensemble of erosional and sediment transport processes have reached an equilibrium at the scale of individual clasts". It's not clear what this actually means.

Yes, this was unclear and has been removed.

Lines 429-430. "the clast fabric of the channel fill has dynamically adjusted to water and sediment flux and their specific timescales". This is also unclear. What timescales are being referred to here? What does it mean to say a "clast fabric has dynamically adjusted"?

We have removed this part.

Fig. 1. The latitude ticks are horizontal while the maps are actually rotated with respect to north, and I'm struggling to identify the latitudes of the catchments. The northernmost catchment looks to be around 9 °S on these maps, but plots at about 7 °S in Fig. 3. The coordinate system of the map could be clearer.

This has been shown in a clearer way on the figure 1

Fig. 1C. Around line 132 the authors describe a major N-S gradient in rainfall rates of ~1000 mm/yr. I can't see evidence for this on the map, and the rainfall rates in the catchments look very uniform. If there is a major N-S gradient it's hidden in the colour scale – consider using a different colour scheme that shows more variation between 0-1000 mm/yr. Also, the caption misspells "extent".

There is a clear E-W trend but no clear N-S trend. The caption has been corrected

Fig. 2. The figure caption has an open bracket.

This has been corrected

Fig. 3. It's unhelpful that the town names don't match Fig. 1.

We have add the name of the city Camana in the figure 1

[3] General suggestions:
Here I am making some general suggestions to the authors, which they can take or leave as they like. I think a better way to interpret the grain size data would be to start with the D50 record in Fig. 3h and look at the spatial patterns. Most of the catchments have a uniform D50 of 2-3mm, and there are two places where there are peaks above this baseline. One is around 11-12 °S, and these catchments are the smaller ones between Lima and Huaraz that don't seem to cross the western escarpment, while the others do. I suggested looking at this in my original review, and it still seems to me that this particular peak in grain size could be related to the catchments being shorter and steeper and not crossing the escarpment, but I'm not familiar enough with the area to develop this further. The authors should think about it, also because these catchments lie right in the centre of the long-wavelength feature visible in the slopes (Fig. 3g). If slopes are mediating grain size via a threshold effect, this could explain why the grain size peak is quite narrow and restricted to only the zone with the highest slopes and these shorter catchments in exactly this location.

The second peak is around 16-17 °S and coincides with a big spike in the mean annual water discharge. It could be that the correlation coefficients don't show a relationship between grain size and water discharge because the authors are plotting loads of noise against itself (most of the catchments have a baseline discharge of 10-20 m3/s), but if they look across latitudes there seems to be an obvious response here. Where discharge jumps up to ~80 m3/s, D50 jumps up to ~6 mm.

Examining these two features in the grain size data would be a better way forward. They are similar in amplitude (D50 increasing by a factor of 2x to 3x, which is significant), but presumably result from different triggers – one to do with discharge (the zone where discharge spikes is where the rivers move coarser material), and the other potentially due to the catchments shortening, not crossing the escarpment, and having the greatest slopes. Both of these responses (to discharge and slopes) could be non-linear and involve thresholds, so the authors need to think about whether simple correlation coefficients are suitable for exploring this (I think not), and whether the relationships get obscured by just doing bulk correlations between all the other catchments as well, when actually the grain size responses are limited to just a few catchments.

If this is correct, then you have two cases where grain size is similarly perturbed (perhaps by thresholds), but as a result of very different perturbations. That's important, because many studies use grain size perturbations to infer climatic/tectonic forcings, but this would suggest both can have similar effects in the sedimentary record that might be difficult to tell apart. Furthermore, the positions of these grain size peaks make sense, suggesting that this is a robust data set that has been measured with great care in the field. The authors have done a great job putting the data set together, now they need to write a paper that does all their hard work justice.

We are very grateful for this comment and have changed the paper accordingly, thereby following closely these recommendations. Our previous version was indeed an overstating of the relationships between week correlations.
We have proceeded carefully with the acquisition of the grain size data and we are happy to read that the Referee considers the data as valuable contribution for the community.
We hope this new version of the manuscript will indeed give justice to the work, which has been done.

[revised manuscript text omitted]

in response to subduction of the oceanic Nazca plate beneath the continental South American plate at least since late Jurassic times (Isacks, 1988). Therefore, it is not surprising that erosion and the transfer of material from the hillslopes to the rivers has been considered to strongly depend on the occurrence of earthquakes, as measured [10]Be concentrations in pebbles suggest (McPhilipps et al., 2014). On the other hand, it has also been proposed that denudation in this part of the Andes is controlled by the distinct N-S and E-W precipitation rate gradients. These inferences have been made based on concentrations of in-situ cosmogenic [10]Be measured in river-born quartz (Abbühl et al., 2011; Carretier et al., 2015; Reber et al., in press 2017), and on morphometric analyses of the western Andean landscape (Montgomery et al., 2001).

BecauseAccordingly, erosion along the western Peruvian Andes has been related to either the occurrence of earthquakes and thus to tectonic processes (McPhillips et al., 2014) or rainfall rates (Abbühl et al., 2011; Carretier et al., 2015) and thus to the stream's mean annual runoff (Reber et al., in press), and since 2017). Therefore, we hypothesize that hillslope erosion andpaired with the supply of material to trunk streams isrunoff are 
[revised manuscript text omitted]